# The repressor Capicua is a barrier to lung tumor development driven by *Kras/Trp53* mutations

Irene Ballesteros-González[1,2], Iván Hernández-Navas[3,4,5], Oksana Brehey[6], Carmen G Lechuga[6], Marina Salmón[6], Morena Scotece[1,2], Ricardo Velasco-Vicente[1,2], Alejandra A Flores-Gómez[1,2], Antonio Cebriá[7], Lucía Simón-Carrasco [6,8,9], Gerardo Jiménez [10,11], Monica Musteanu[5,12], Carmen Guerra[5,6], Orlando Domínguez[13], Eduardo Caleiras[14], Carmen Blanco-Aparicio [7], Tirso Pons[15], Irene Ferrer[3,4,5], Luis Paz-Ares[5,16,17], Raul Torres-Ruiz [18,19,20,21], Sandra Rodríguez-Perales[18], Mariano Barbacid[5,6] & Matthias Drosten [1,2,5✉]

## Abstract

*KRAS* mutations are responsible for a quarter of all lung adenocarcinomas. However, the molecular mechanisms linking these mutations and their frequent secondary dosage amplification to tumor formation are still not fully understood. While ample evidence supports a crucial role for the MAPK pathway in tumor development, the primary effectors targeted by this pathway remain largely unexplored. Here we identify the transcriptional repressor Capicua (CIC) as a key target inactivated by KRAS/MAPK signaling in lung adenocarcinoma. We show that genetic loss of CIC recapitulates the phenotypic consequences of amplified KRAS signaling. Genetic disruption of CIC suppressed the requirement for *Kras* allelic imbalances and accelerated the transformation of bronchiolar Club cells. We also demonstrate that restoring CIC repressor activity impaired proliferation of CIC-deficient tumor cells and reverted resistance to MAPK pathway inhibitors. These results highlight the key role of CIC during lung tumor formation and suggest that selective pressure for effective CIC inactivation favors secondary amplification of KRAS/MAPK signaling in tumor cells.

**Key words** KRAS; Lung Cancer; Allelic Imbalance; Drug Resistance; Repression
**Subject Categories** Cancer; Respiratory System

## Introduction

Lung cancer is one of the most frequently diagnosed cancers and the leading cause of cancer-related mortality worldwide. Mutations in *KRAS* contribute to ~25% of lung adenocarcinomas (LUADs), the most frequent lung cancer subtype (Ferrer et al, 2018; Sung et al, 2021). Although initially classified as undruggable, there are now two FDA-approved drugs targeting a specific mutant isoform (KRAS[G12C]). This has been the starting point for the development of a wide variety of other allele-specific, pan-KRAS, or even pan-RAS inhibitors (Holderfield et al, 2024), although the frequent emergence of resistance, at least in the case of KRAS[G12C] inhibitors, has not yet produced major clinical benefits (Drosten and Barbacid, 2022; Zhu et al, 2022; de Langen et al, 2023; Molina-Arcas and Downward, 2024).

Despite these advances, surprisingly little is known about the molecular mechanisms linking *KRAS* mutations to lung cancer. In mice, only a small fraction of cells expressing a resident *Kras* oncogene progress to form tumors, suggesting that additional mechanisms must contribute to tumor growth (Guerra et al, 2003; Mainardi et al, 2014). Whole-exome sequencing of lung tumors

[1]Molecular Mechanisms of Cancer Program, Centro de Investigación del Cáncer (CIC), 37007 Salamanca, Spain. [2]Instituto de Biología Molecular y Celular del Cáncer (IBMCC), CSIC-USAL, 37007 Salamanca, Spain. [3]Grupo de Terapias Dirigidas para la Oncología de Precisión & Unidad de Investigación Clínica de Cáncer de Pulmón, Instituto de Investigación Hospital 12 de Octubre, 28041 Madrid, Spain. [4]Clinical Research Program, Centro Nacional de Investigaciones Oncológicas (CNIO), 28029 Madrid, Spain. [5]Centro de Investigación Biomédica en Red de Cáncer (CIBERONC), Instituto de Salud Carlos III, 28029 Madrid, Spain. [6]Tumor Biology Program, Centro Nacional de Investigaciones Oncológicas (CNIO), 28029 Madrid, Spain. [7]Experimental Therapeutics Program, Centro Nacional de Investigaciones Oncológicas (CNIO), 28029 Madrid, Spain. [8]Centro Andaluz de Biología Molecular y Medicina Regenerativa (CABIMER), Consejo Superior de Investigaciones Científicas (CSIC), Universidad de Sevilla, 41092 Sevilla, Spain. [9]Universidad Pablo de Olavide, 41013 Sevilla, Spain. [10]Instituto de Biología Molecular de Barcelona (IBMB), CSIC, 08028 Barcelona, Spain. [11]Institució Catalana de Recerca i Estudis Avançats (ICREA), 08010 Barcelona, Spain. [12]Department of Biochemistry and Molecular Biology, Faculty of Pharmacy, Complutense University of Madrid, 28040 Madrid, Spain. [13]Genomics Unit, Centro Nacional de Investigaciones Oncológicas (CNIO), 28029 Madrid, Spain. [14]Histopathology Unit, Centro Nacional de Investigaciones Oncológicas (CNIO), 28029 Madrid, Spain. [15]Department of Immunology and Oncology, National Center for Biotechnology (CNB-CSIC), Spanish National Research Council, 28049 Madrid, Spain. [16]Unidad de Investigación Clínica de Cáncer de Pulmón, Instituto de Investigación Hospital 12 de Octubre, 28041 Madrid, Spain. [17]Complutense University of Madrid, 28040 Madrid, Spain. [18]Molecular Cytogenetics and Genome Editing Unit, Centro Nacional de Investigaciones Oncológicas (CNIO), 28029 Madrid, Spain. [19]Division of Hematopoietic Innovative Therapies, Biomedical Innovation Unit, Centro de Investigaciones Energéticas, Medioambientales, y Tecnológicas (CIEMAT), 28040 Madrid, Spain. [20]Advanced Therapies Unit, Instituto de Investigación Sanitaria Fundación Jiménez Díaz, 28003 Madrid, Spain. [21]Centro de Investigación Biomédica en Red de Enfermedades Raras (CIBERER), 28029 Madrid, Spain. ✉E-mail: mdrosten@usal.es

driven by endogenous *Kras* oncogenes has revealed a relatively modest number of co-occurring mutations, including a significant fraction of tumors lacking additional mutations in other known cancer genes (Chung et al, 2017; Junttila et al, 2010; McFadden et al, 2016; Westcott et al, 2015). In contrast, most of these tumors carried secondary amplifications of the *Kras* locus. Importantly, human *KRAS*-mutant LUADs also show frequent allelic imbalances associated with reduced patient survival (Chiosea et al, 2011; Yu et al, 2017). Moreover, it has been shown that *Kras* allelic imbalance promotes malignancy, at least in part, through metabolic rewiring and stimulation of tumor cell fitness (Burgess et al, 2017; Kerr et al, 2016). Consistent with the requirement for MAPK signaling in KRAS-driven lung cancer, these findings suggest that *Kras* allelic imbalance may further promote MAPK pathway activity and tumor growth (Blasco et al, 2011; Burgess et al, 2017; Drosten and Barbacid, 2020; Junttila et al, 2010). While gaining additional copies of mutant *Kras* alleles is particularly frequent in LUAD (Chung et al, 2017; McFadden et al, 2016; Westcott et al, 2015), concomitant loss of the wild-type allele (loss of heterozygosity) can also enhance cellular transformation and MAPK pathway activity (Ambrogio et al, 2018).

The transcriptional repressor Capicua (CIC) has been identified as a critical substrate of ERK kinases in development and disease (Jiménez et al, 2012; Kim et al, 2021; Lee, 2020; Simón-Carrasco et al, 2018; Wong and Yip, 2020). When ERK kinases are inactive, CIC proteins engage highly conserved octameric DNA binding sites and repress transcription of their target genes by recruiting the SIN3 deacetylation complex (Ajuria et al, 2011; Weissmann et al, 2018; Gupta et al, 2022). In contrast, ERK activation results in CIC phosphorylation and inactivation of its repressor activity, thereby leading to derepression of its targets (Bunda et al, 2019; Dissanayake et al, 2011; Okimoto et al, 2017; Park et al, 2022).

Accumulating evidence supports the view that *CIC* is a tumor suppressor in human cancer (Kim et al, 2021). *CIC* mutations have been detected at high frequencies in oligodendrogliomas, and several other tumor types, including stomach adenocarcinoma, melanoma or lung cancer, also show significant mutation rates (Kim et al, 2021; Simón-Carrasco et al, 2018). Moreover, increased proteasomal degradation of CIC has been observed in glioblastomas (Bunda et al, 2019), and homozygous deletions of *CIC* occur in a variety of cancers (Kim et al, 2021). Of note, *CIC* deletions as well as inactivating mutations were associated with metastasis of lung cancer cells (Okimoto et al, 2017).

In addition, *CIC* mutations have been linked to resistance to drugs targeting RTK/RAS/MAPK signaling (Simón-Carrasco et al, 2017; Kim et al, 2022; Wang et al, 2017; Liao et al, 2017). Indeed, inactivating mutations were identified as the principal mechanism of resistance in patients treated with MAPK pathway inhibitors (Hashiba et al, 2020; Da Vià et al, 2020). These observations suggest that CIC is a critical effector of the RTK/RAS/MAPK signaling pathway in cancer development and therapy resistance, although its contribution to KRAS-driven LUAD has remained unknown. Hence, in this study, we have interrogated the role of CIC in LUAD formation and drug resistance using genetically-engineered mouse (GEM) models as well as patient-derived organoids, and reveal that CIC inactivation enhances tumor initiation. We also show that, while promoting resistance to inhibitors of the MAPK pathway, absence of CIC creates exploitable vulnerabilities in these tumors.

# Results

## Genetic CIC inactivation facilitates lung tumor initiation

LUADs driven by *Kras* oncogenes exhibit stage-specific amplification of MAPK signaling (Chung et al, 2017; Cicchini et al, 2017; Chen et al, 2019; Feldser et al, 2010). Thus, we hypothesized that amplified MAPK signaling may be relevant for efficient suppression of growth-inhibitory activities that prevent uncontrolled tumor progression, such as those mediated by ERK-regulated transcriptional repressors. Based on this assumption, we embarked on exploring the role of the repressor Capicua (CIC), which is negatively controlled by active MAPK signaling (Simón-Carrasco et al, 2018). To this end, we bred $Cic^{lox/lox}$ mice (Simón-Carrasco et al, 2017) with the $Kras^{+/LSLG12Vgeo}$;$Trp53^{lox/lox}$ (KP) strain known to develop aggressive lung adenocarcinomas upon Cre-mediated recombination (Fig. 1A) (Drosten et al, 2017). In these mice, CIC can be rendered inactive by eliminating sequences corresponding to exons 2–6, resulting in the expression of non-functional CIC-S$^{Δ2-6}$ and CIC-L$^{Δ2-6}$ protein isoforms that lack their HMG-box DNA binding domain required for gene repression (Simón-Carrasco et al, 2017). Intranasal infection of KP mice with Ad-Cre resulted in the death of all animals due to lung tumor development with a median survival of 48 weeks (Fig. 1B). Notably, infection of $Kras^{+/LSLG12Vgeo}$;$Trp53^{lox/lox}$;$Cic^{lox/lox}$ (KPCic) mice resulted in significantly reduced survival (median survival 39 weeks) (Fig. 1B). As illustrated in Fig. EV1A,B, KPCic mice displayed a significant increase in grade 3 adenocarcinomas when sacrificed at a humane endpoint (Jackson et al, 2005). However, we did not observe evidence for accelerated tumor progression or a shift towards more aggressive lesions in the absence of functional CIC, as demonstrated by HMGA2 immunostaining as a marker for lung tumor progression and invasion (Fig. EV1C–E) (Winslow et al, 2011).

To ascertain whether the reduced survival of KPCic mice stemmed from more efficient initiation of lung tumors, we infected KPCic as well as KP mice as controls with Ad-Cre and compared their tumor burden 20 weeks post-infection. At this time point, KPCic mice exhibited an approximately 2.5-fold increase in the number of lesions (Fig. 1C). Yet, tumors in KPCic mice were histologically indistinguishable from those in KP mice, both expressing TTF-1 (also known as NKX2-1) and the alveolar marker SPC, but lacking the marker of bronchiolar Club cells CC10 (Fig. 1D). As shown in Fig. EV1F, the overall increase in the number of lesions in KPCic mice resulted from an elevated number of grade 2 adenomas and grade 3 adenocarcinomas. The percentage of HMGA2-positive tumors was similar in both groups, again supporting the idea that inactivation of CIC had no impact on tumor progression (Fig. EV1G,H). Interestingly, tumors from KPCic mice displayed significantly lower levels of pERK+ areas, while the percentage of Ki67+ cells was similar in both groups (Fig. 1E,F).

To better understand the impact of *Cic* disruption on tumor growth in KP mice, we first determined the genes potentially controlled by CIC by ChIP sequencing (ChIP-seq) in KPCic cells vs. KP cells left untreated or treated with the selective MEK inhibitor trametinib for 24 h to efficiently block the MAPK pathway and maximize CIC DNA binding. While the inactive CIC$^{Δ2-6}$ protein in KPCic cells, as expected (Simón-Carrasco et al, 2017), did not bind to CIC target genes (29% of peaks were located in

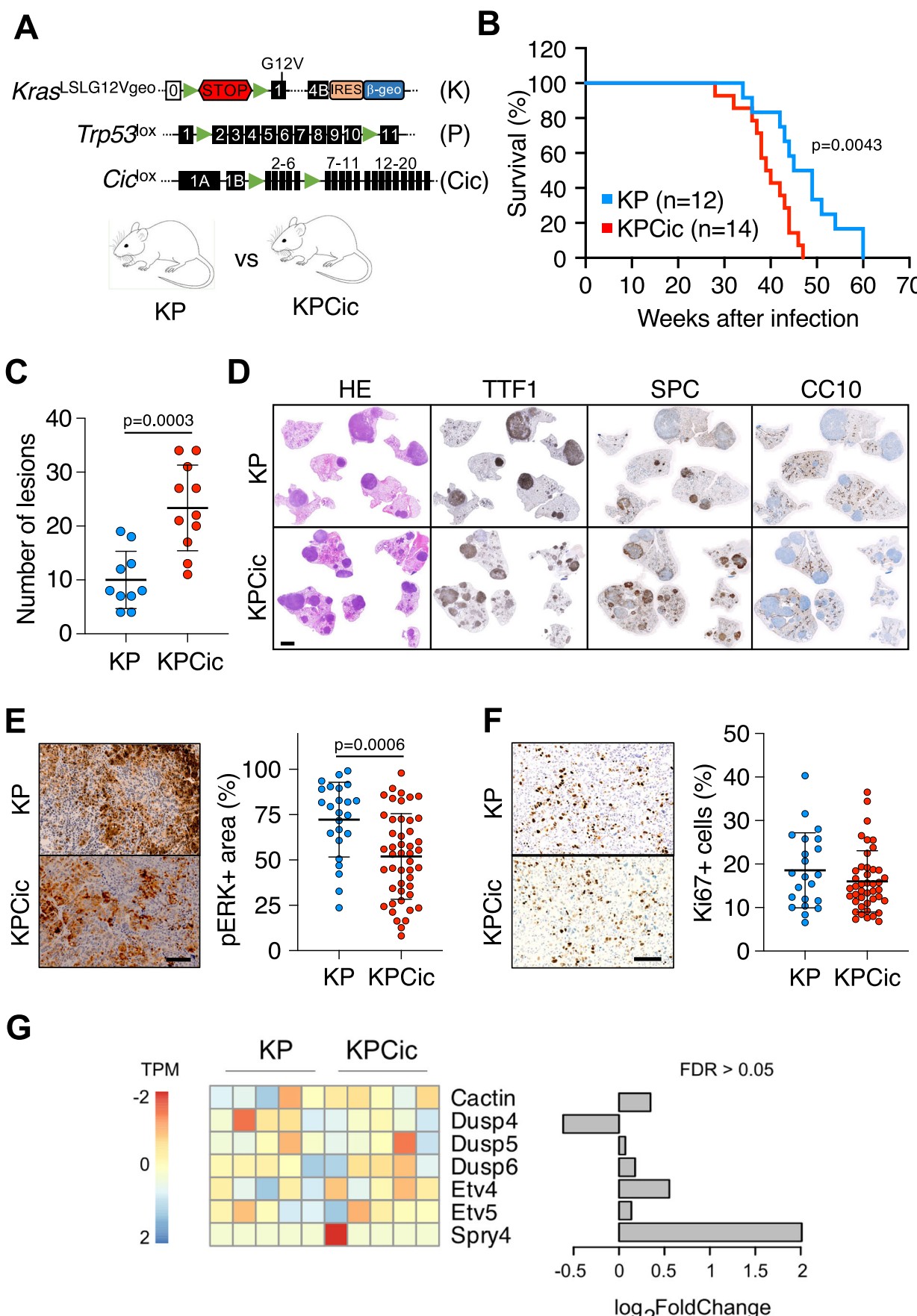

**Figure 1. Genetic inactivation of CIC promotes lung tumor initiation driven by *Kras* oncogenes.**

(A) Schematic representation of mouse strains used in this study. KP, *Kras*$^{+/LSLG12Vgeo}$;*Trp53*$^{lox/lox}$. KPCic, *Kras*$^{+/LSLG12Vgeo}$;*Trp53*$^{lox/lox}$;*Cic*$^{lox/lox}$. Coding and non-coding exons are represented by black and white boxes, respectively. The transcriptional termination sequence (STOP) is shown as a red hexagon, the internal ribosome entry site (IRES) as a brown rectangle and the *LacZ/Neo*$^R$ fusion gene (β-geo) in blue. *LoxP* sites are demonstrated as green triangles. (B) Survival of KP (blue, $n = 12$) and KPCic mice (red, $n = 14$) infected with $5 \times 10^5$ pfu Ad-Cre. Statistics, log-rank test. (C) Number of lesions in KP (blue, $n = 10$ mice) and KPCic (red, $n = 11$ mice) mice 5 months after infection with $5 \times 10^7$ pfu Ad-Cre. Results are shown as mean ± SD. Statistics, unpaired *t*-test. (D) Representative images of hematoxylin & eosin staining (HE), as well as TTF-1 (NKX2-1), SPC and CC10 immunostaining in paraffin-embedded sections of lungs from KP and KPCic mice 5 months after infection with Ad-Cre. Scale bar, 2 mm. (E) Left: Representative images of pERK immunostaining in tumor sections from KP and KPCic mice 5 months after infection with Ad-Cre. Scale bar, 100 μm. Right: Quantification of pERK+ areas in tumors from KP (blue, $n = 24$ tumors) or KPCic mice (red, $n = 47$ tumors) 5 months after infection with Ad-Cre. Results are shown as mean ± SD. Statistics, unpaired *t*-test. (F) Left: Representative images of Ki67 immunostaining in tumor sections from KP and KPCic mice 5 months after infection with Adeno-Cre. Scale bar, 100 μm. Right: Quantification of Ki67+ cells in tumors from KP (blue, $n = 22$ tumors) or KPCic mice (red, $n = 43$ tumors) 5 months after infection with Ad-Cre. Results are shown as mean ± SD. Statistics, unpaired *t*-test. (G) Heatmap of expression of selected CIC target genes and their respective Log$_2$FoldChange values in tumors from KP or KPCic mice as determined by RNA-seq. Source data are available online for this figure.

other promoter regions based on ChIPseeker annotations), CIC proteins in proliferating KP cells showed a significant association with several promoters (69% of all peaks) including known CIC targets (Fig. EV2 and Dataset EV1). However, upon trametinib treatment, CIC strongly relocated to a more limited set of promoters (44% of all peaks), including *Etv4* and *Etv5*, two well-known CIC target genes (Kawamura-Saito et al, 2006), or well-known negative regulators of the MAPK pathway such as *Spry2*, *Dusp4*, *Dusp5* and *Dusp6* (Fig. EV2A,B and Dataset EV1). Yet, expression of most genes, including these CIC targets, was unaltered in tumors from KPCic mice, suggesting that once tumors have formed in the absence of CIC, they are largely indistinguishable from those in KP mice (Fig. 1G, Fig. EV2C and Dataset EV2). Together, these findings indicate that CIC inactivation facilitates lung tumor initiation but does not notably contribute to tumor progression in KP mice.

## Genetic CIC inactivation abrogates *Kras* allelic imbalance

Similar to human tumors, GEM models of KRAS-driven LUAD frequently develop allelic imbalance via acquisition of additional copies of mutant *Kras* alleles (Chung et al, 2017; McFadden et al, 2016; Westcott et al, 2015). This suggests that such allelic expansions could be, at least to some extent, responsible for amplified MAPK signaling in those tumors. Indeed, we noted that allelic imbalance increased significantly over time in KP mice infected with Ad-Cre as quantified by fluorescence-in situ hybridization (FISH) (Fig. 2A,B). Moreover, MAPK signaling increased concomitantly over time as demonstrated by pERK+ immunostaining (Fig. 2A,C).

Thus, we hypothesized that *Kras* allelic imbalance and amplified MAPK signaling could represent a mechanism selected to ensure effective inactivation of CIC and thus facilitate tumor initiation. In this view, CIC repressor activity would act as a barrier to LUAD development, while *Kras* allelic amplification would promote sufficient inactivation of CIC to overcome this barrier. However, as we examined this idea, we unexpectedly found that LUAD tumors harboring excess MAPK activity did not present changes in CIC protein levels or localization (Fig. 2A). Similarly, using our recently described *Kras*$^{+/G12Vlox}$;*Trp53*$^{-/-}$;*Rosa26-CreERT2*$^{KI/KI}$;*hUBC-CreERT2*$^{+/T}$ (K$^{G12Vlox}$PC2) cell lines, in which the oncogenic *Kras*$^{G12V}$ allele can be excised by adding 4-hydroxytamoxifen (Salmón et al, 2023), we observed that elimination of *Kras*$^{G12V}$ had little impact on localization or stability of CIC (Fig. EV3A,B). However, LUAD tumors do show a significant increase in ETV5+ cells (Fig. 2A,D),

raising the possibility that amplified MAPK signaling might inhibit CIC repressor activity by a mechanism independent of subcellular localization of protein degradation. Indeed, loss of *Kras*$^{G12V}$ in K$^{G12Vlox}$PC2 cells resulted in enhanced binding of CIC to its target promoters, suggesting that CIC activity in LUAD is regulated at the level of DNA binding (Fig. EV3C). Notably, we also found that genetic disruption of CIC significantly suppressed the mutant *Kras* allelic expansion characteristic of KP tumors (Fig. 2E,F). However, this suppression was not as evident in tumors retaining functional p53 (Fig. EV4A,B). Since LUADs also show a frequent gradual increase in ERBB activity (Kruspig et al, 2018), we also analyzed whether the absence of CIC affected expression of ERBB family receptors and ligands. As shown in Fig. EV4C, we did not detect differences in their expression levels in KP and KPCic tumors, suggesting that this mechanism of tumor progression is not affected by the absence of CIC, at least at the time point analyzed. Interestingly, in human LUAD as well as other tumor types carrying *KRAS* mutations, the presence of *CIC* mutations correlates with the lack of *KRAS* oncogene amplification (Fig. EV4D,E). Together, these observations indicate that CIC inactivation, either via mutation or in response to sufficiently strong, amplified MAPK signaling, is key for LUAD development in KP mice.

## Genetic CIC inactivation promotes expansion and lineage conversion of Club cells

Next, we aimed at identifying the origin of the elevated tumor burden observed upon genetic disruption of *Cic*. To this end, we infected KP as well as KPCic mice with Ad-Cre and quantified clusters of X-Gal+ cells as a surrogate marker for KRAS$^{G12V}$ expression 4 weeks post-infection (Guerra et al, 2003; Mainardi et al, 2014). As illustrated in Fig. 3A,B, we did not observe differences in the number and size of X-Gal+ clusters between KP and KPCic mice within alveoli. However, KPCic mice exhibited significantly more X-Gal+ clusters with 5 or more cells within or close to bronchioles, suggesting that, despite the absence of the bronchiolar marker CC10 in full-blown tumors, CIC inactivation facilitated expansion of bronchiolar cells. Indeed, accumulating evidence indicates that bronchiolar Club cells can serve as the cellular origin for KRAS-driven lung tumors while transitioning from expressing bronchiolar to alveolar lineage markers (Rosigkeit et al, 2021; Spella et al, 2019; Cicchini et al, 2017; Sutherland et al, 2014; Nieto et al, 2017). To ascertain whether additional tumors arising in KPCic mice originated from Club cells, we infected KP and KPCic mice with Ad-CC10-Cre particles that restrict Cre

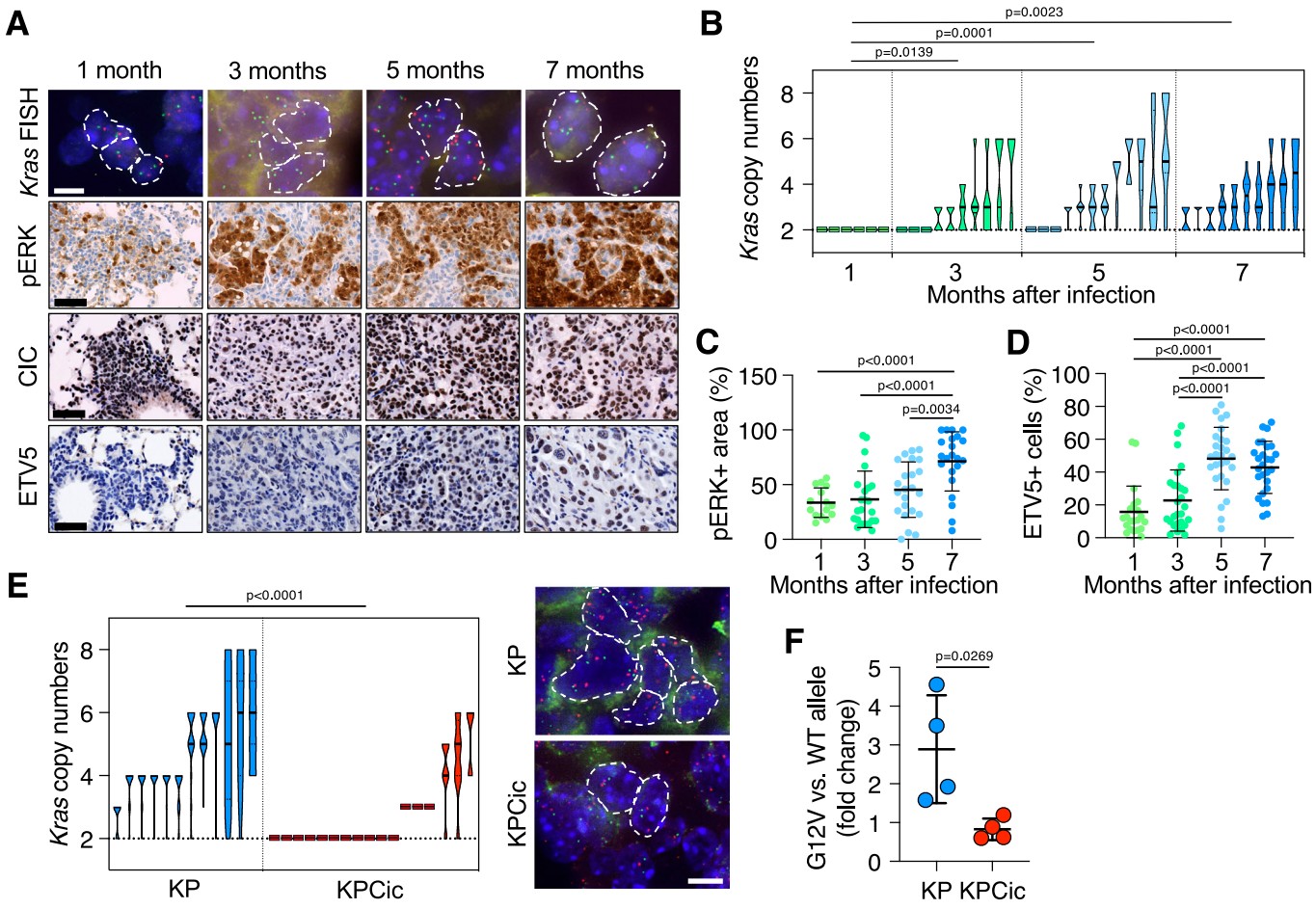

**Figure 2. Genetic CIC inactivation abrogates the requirement for Kras allelic imbalance**

(A) Representative images of Kras interphase FISH as well as pERK, CIC, and ETV5 immunostaining in sections from lung tumors in KP and KPCic mice 5 months after infection with Ad-Cre. FISH images: Red, Kras probe. Green, chromosome 6 probe. Scale bars, 5 μm (FISH) or 50 μm (immunostaining). Dotted lines indicate cell borders. (B) Quantification of Kras copy numbers by FISH in sections from lung tumors in KP mice obtained 1 ($n = 6$ tumors; $n = 2$ mice), 3 ($n = 10$ tumors; $n = 2$ mice), 5 ($n = 12$ tumors, $n = 3$ mice) and 7 months ($n = 10$ tumors; $n = 2$ mice) after infection with Ad-Cre. Statistics, $X^2$ tests. (C) Quantification of pERK+ areas in tumors from KP mice obtained at the indicated time points after infection with Ad-Cre (1 month, $n = 15$ tumors; 3 months, $n = 23$ tumors; 5 months, $n = 23$ tumors; 7 months, $n = 23$ tumors). Statistics, one-way ANOVA with Tukey's multiple comparison test. (D) Quantification of ETV5+ cells in tumors from KP mice obtained at the indicated time points after infection with Ad-Cre (1 month, $n = 21$ tumors; 3 months, $n = 27$ tumors; 5 months, $n = 27$ tumors; 7 months, $n = 27$ tumors). Statistics, one-way ANOVA with Tukey's multiple comparisons test. (E) Left, quantification of Kras copy numbers by FISH staining in sections from lung tumors in KP ($n = 12$ tumors, $n = 3$ mice; blue symbols) and KPCic ($n = 17$ tumors, $n = 3$ mice, red symbols) mice obtained 5 months after infection with Ad-Cre. Statistics, $X^2$ tests. Right, representative interphase FISH images of tumors obtained from KP and KPCic mice 5 months after infection with Ad-Cre. Red, Kras probe. Green, chromosome 6 probe. Scale bar, 5 μm. Dotted lines indicate cell borders. (F) Relative levels of Kras alleles determined by semiquantitative PCR in lung tumors obtained from KP ($n = 4$) and KPCic ($n = 4$) mice 5 months after infection with Ad-Cre. Results are shown as mean ± SD. Statistics, unpaired t-test. Source data are available online for this figure.

expression to CC10-positive cells such as Club cells (Sutherland et al, 2011). Immunostaining of CC10 and SPC markers revealed the emergence of double-positive cells within bronchioles 4 weeks post-infection in both KP and KPCic mice, indicating that activation of KRAS^{G12V} with concomitant elimination of p53 caused a subset of Club cells to express alveolar type 2 (AT2) cell markers (Fig. 3C). Yet, these cells expanded more rapidly in KPCic mice as shown by the appearance of significantly larger clusters of double-positive cells that ceased to express CC10 when they continued to grow, while mostly retaining expression of SOX2, another Club cell marker (Sutherland et al, 2014) (Figs. 3C,D and EV4F). Importantly, KPCic mice also displayed significantly more tumors 4 months after infection with Ad-CC10-Cre, indicating that accelerated conversion of Club cells into AT2-marker-positive

tumor cells and their subsequent expansion could be a possible explanation for their increased tumor burden (Fig. EV4G).

## Latent Kras mutations cooperate with concomitant loss of Trp53 and Cic in tumor initiation

The above results underscore the contribution of CIC inactivation to lung tumor development driven by Kras oncogenes. Although we previously observed that systemic disruption of Cic in mice for up to 9 months did not result in any histopathological alteration in lung tissue (Simón-Carrasco et al, 2017), we now explored whether latent mutations in other oncogenes might cooperate with the disruption of Cic in lung tumor development. Interestingly, two thirds of human CIC-mutant LUADs harbor co-occurring mutations in TP53 or other

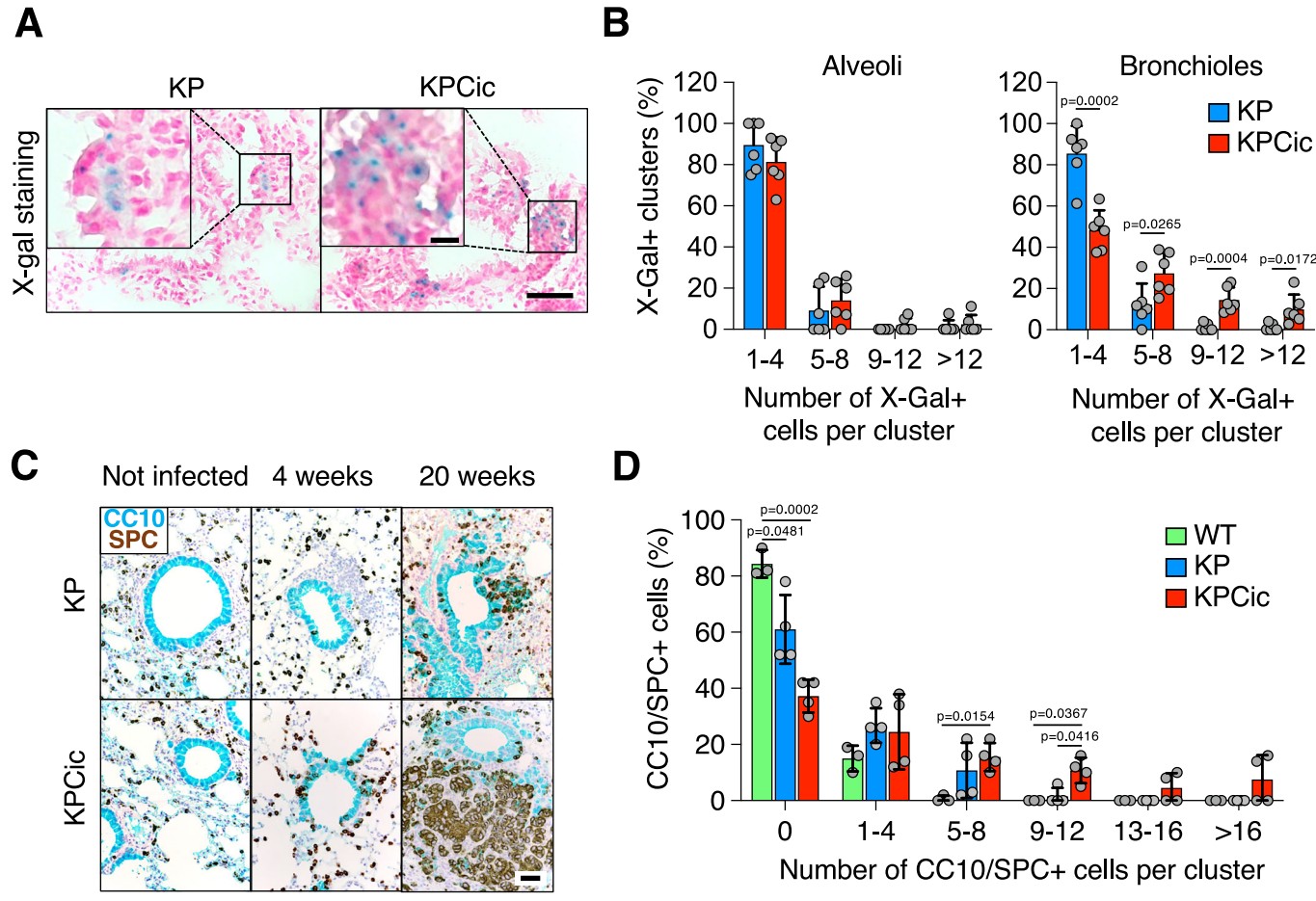

**Figure 3. Genetic CIC inactivation promotes expansion and lineage conversion of Club cells.**

(A) Representative images of X-Gal staining in lung sections from KP and KPCic mice 4 weeks after infection with Ad-Cre. Scale bar, 50 μm (insets, 10 μm). (B) Percentage of X-Gal+ cell clusters in lung sections from KP (blue, n = 6) and KPCic mice (red, n = 6) 4 weeks after infection with Ad-Cre. Results are shown as mean ± SD. Statistics, multiple t-tests. (C) Representative images of SPC (brown) and CC10 (blue) staining in lung sections from KP and KPCic mice, either left untreated, or 4 and 20 weeks after infection with Ad-CC10-Cre. Scale bar, 50 μm. (D) Percentage of SPC/CC10-double-positive cell clusters in WT (green, n = 3), KP (blue, n = 4) or KPCic (red, n = 4) mice 4 weeks after infection with Ad-CC10-Cre. Results are shown as mean ± SD. Statistics, two-way ANOVA with Tukey's multiple comparison test. Source data are available online for this figure.

genes mutated in lung cancer, including *KRAS* (Fig. 4A). Thus, we infected *Cic*^lox/lox^, *Trp53*^lox/lox^ or *Trp53*^lox/lox^;*Cic*^lox/lox^ (PCic) mice with Ad-Cre and evaluated their tumor burden after 1 year. Although none of these animals became sick during this time, further histopathological analyses revealed the presence of lung adenocarcinomas positive for TTF-1 and SPC in 10 out of 22 PCic mice (Fig. 4B–D). Interestingly, neither *Cic*^lox/lox^ nor *Trp53*^lox/lox^ mice infected with Ad-Cre showed histological alterations in their lungs at the same time point. Further characterization of these tumors by whole exome sequencing revealed the presence of latent or spontaneous *Kras*^Q61R^ mutations in 3/3 tumors analyzed (Fig. 4E). Moreover, these mutations were present at low frequencies, suggesting that none of these tumors showed notable *Kras* oncogene amplifications, as typically observed in KP mice. These tumors also had significantly lower levels of pERK than those from KP mice when sacrificed at a humane endpoint (Fig. 4F). A fraction of *KRAS/TP53* mutant LUAD patient samples in The Cancer Genome Atlas (TCGA) database also harbor loss-of-function mutations in *CIC* that lack *KRAS* imbalances (Appendix Fig. S1A). Interestingly, these tumors

show no differences in the expression of a genetic signature indicative of MAPK pathway activation that includes multiple CIC target genes (Pratilas et al, 2009), consistent with our findings in mice (Appendix Fig. S1B). Yet, *KRAS/TP53/CIC*-mutant LUADs display multiple differentially expressed genes when compared with *KRAS/TP53*-mutant tumors, although this, as well as the lack of allelic imbalances, needs to be confirmed in additional tumor samples (Appendix Fig. S1C). Moreover, since the number of tumors with this mutational profile is relatively low, it remains to be determined whether *CIC* mutations influence any clinical parameters. Taken together, our data suggest that disruption of *Cic* facilitates initiation of lung adenocarcinomas in KP mice, possibly by rendering *Kras* allelic expansion unnecessary at early stages.

## Reintroduction of CIC decreases tumor cell proliferation and reverts resistance to MAPK pathway inhibitors

The above results underscore the impact of CIC inactivation in KRAS-driven LUAD. Hence, we aimed at exploring whether

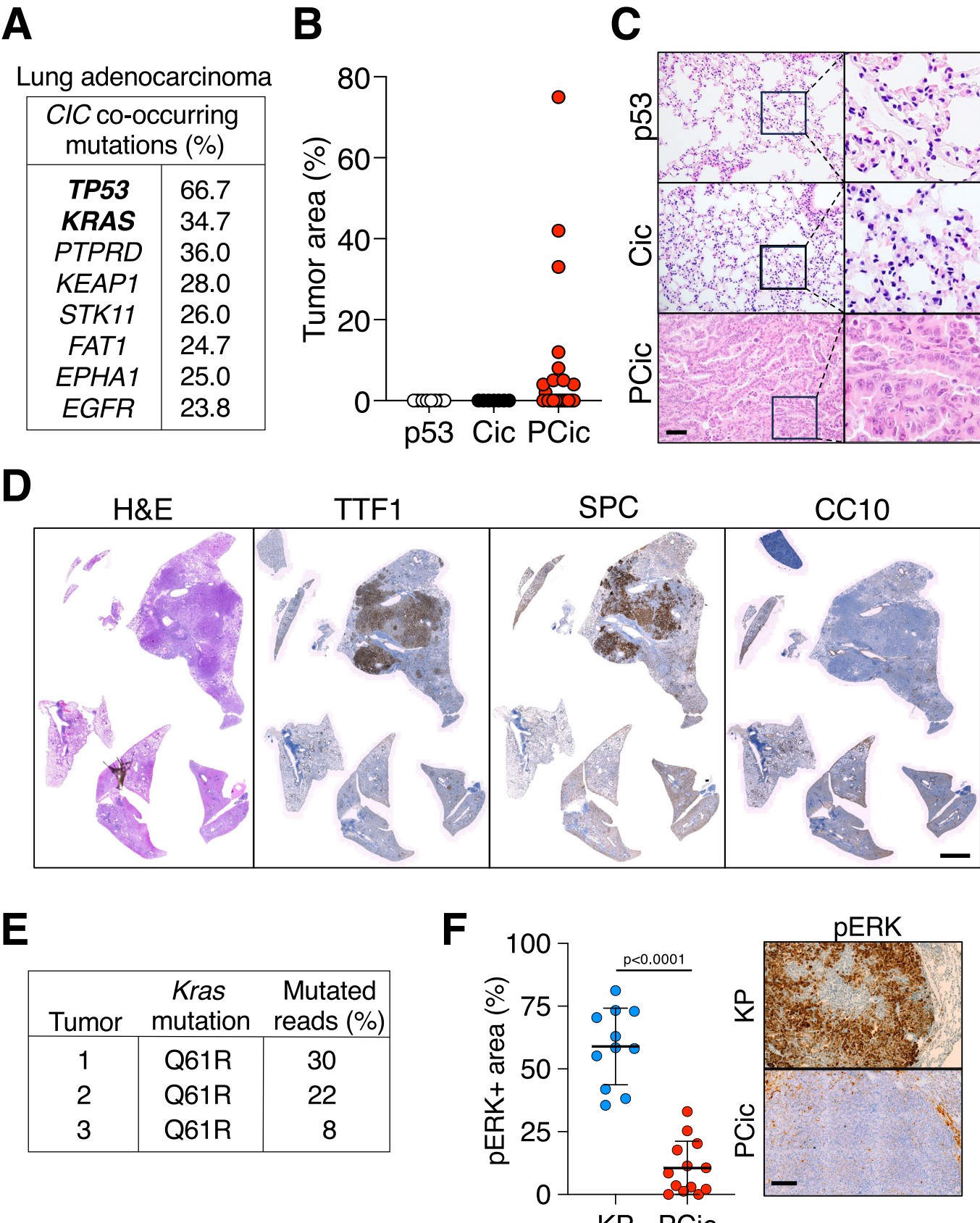

**A**

Lung adenocarcinoma

| *CIC* co-occurring mutations (%) | |
|---|---|
| ***TP53*** | 66.7 |
| ***KRAS*** | 34.7 |
| *PTPRD* | 36.0 |
| *KEAP1* | 28.0 |
| *STK11* | 26.0 |
| *FAT1* | 24.7 |
| *EPHA1* | 25.0 |
| *EGFR* | 23.8 |

**B**

**C**

**D**

H&E    TTF1    SPC    CC10

**E**

| Tumor | *Kras* mutation | Mutated reads (%) |
|---|---|---|
| 1 | Q61R | 30 |
| 2 | Q61R | 22 |
| 3 | Q61R | 8 |

**F**

pERK

**Figure 4. Genetic inactivation of CIC drives lung tumor formation in cooperation with *Trp53* loss and latent *Kras* mutations.**

(A) *CIC* co-occurring mutations in human lung adenocarcinomas. Data were obtained from the TCGA database. (B) Percentage of tumor area in lung sections from *Trp53*^lox/lox^ (p53, white circles, n = 8), *Cic*^lox/lox^ (Cic, black circles, n = 7) or *Trp53*^lox/lox^;*Cic*^lox/lox^ (PCic, red circles, n = 22) mice 1 year after infection with Ad-Cre. (C) Representative images of lung sections 1 year after infection with Ad-Cre in *Trp53*^lox/lox^ (p53), *Cic*^lox/lox^ (Cic) or *Trp53*^lox/lox^;*Cic*^lox/lox^ (PCic) mice. Scale bar, 50 μm. (D) Representative images of hematoxylin & eosin (HE) staining as well as TTF-1 (NKX2-1), SPC and CC10 immunostaining of tumors in lung sections from a *Trp53*^lox/lox^;*Cic*^lox/lox^ (PCic) mouse 1 year after infection with Ad-Cre. Scale bar, 2 mm. (E) Identification of latent *Kras* mutations by WES in tumors from PCic mice. (F) Left: Quantification of pERK+ areas in tumors from KP (blue, n = 11 tumors) or PCic mice (red, n = 13 tumors) at their humane endpoint, after infection with Ad-Cre. Results are shown as mean ± SD. Statistics, unpaired *t*-test. Right: Representative images of pERK immunostaining in tumor sections from KP and PCic mice at their humane endpoint after infection with Ad-Cre. Scale bar, 200 μm. Source data are available online for this figure.

reactivation of CIC could represent a potential therapeutic scenario. To this end, we established cell lines from tumors that developed 5 months after infection in KP and KPCic mice. These tumor cell lines did not differ in their growth properties or the activity of classical signaling pathways downstream of KRAS (Figs. 5A and EV5A). Yet, as expected (Simón-Carrasco et al, 2017; Wang et al, 2017; Liao et al, 2017), KPCic cells exhibited resistance to trametinib (Fig. 5B). To test whether reactivation of CIC affected cell proliferation and/or resistance to trametinib, we infected KP and KPCic tumor cells with adenoviral vectors expressing GFP as a control, the CIC-S cDNA fused to GFP (^GFP^CIC) (Dissanayake et al, 2011) or phosphorylation-insensitive ^GFP^CIC^S173A^ known to more effectively repress its target genes (Park et al, 2023). Moreover, to efficiently reconstitute CIC's transcriptional repressor activity, we ectopically expressed the corepressor ATXN1L in these experiments (Fig. 5C) (Lee et al, 2011; Wong et al, 2019), since cell lines established from KP and KPCic tumors displayed significantly lower levels of *Atxn1* and *Atxn1l* than the respective tumors, which were insufficient to allow efficient transcriptional repression upon adenoviral expression of CIC proteins (Fig. EV5B,C). As shown in Fig. 5D, ectopic expression of ^GFP^CIC still caused little or no repression of *Etv4* and *Etv5*. However, ectopic expression of ^GFP^CIC^S173A^ resulted in efficient repression of these genes in most KP and KPCic cells. Likewise, ^GFP^CIC^S173A^ was also more effective than ^GFP^CIC in preventing colony growth of KPCic cells, while both CIC variants were equally effective in KP cells (Fig. 5E,F; Appendix Fig. S2). More importantly, however, expression of ^GFP^CIC and ^GFP^CIC^S173A^ re-sensitized resistant KPCic cells to trametinib treatment (Fig. 5F and Appendix Fig. S2).

## ETV4 and ETV5 mediate proliferation and resistance to MAPK pathway inhibition

Next, we examined which CIC-controlled genes contribute to KRAS-driven tumor cell proliferation and drug resistance. To this end, we conducted RNA-seq of three independent KP and KPCic tumor cell lines either treated with 20 nM trametinib for 24 h or DMSO as a control. We reasoned that while trametinib treatment would inhibit the MAPK pathway in KP cell lines, resulting in higher CIC repressor activity, the lack of functional CIC in KPCic cells should prevent repression of its targets. As demonstrated in Fig. 6A; Dataset EV3, the genes that best follow this expression pattern are *Etv4* and *Etv5*. Constitutive derepression of these genes, as well as *Ccnd1*, another known CIC target (Dataset EV1), was also confirmed by quantitative PCR in KPCic cells, while other known MAPK-regulated genes do not follow this pattern (Figs. 6B and EV5D), despite binding of CIC to their promoters (Fig. EV2A,B). These results also indicate that, although endogenous *Atxn1*/*Atxn1l*

levels are significantly reduced, they remain sufficient to sustain CIC repressor activity in KP cell lines without the need for ectopic ATXN1L expression. Interestingly, ETV4 and ETV5 protein levels were not entirely MAPK pathway-independent in KPCic cells, suggesting the existence of CIC-independent mechanisms controlling their translation or stability (Fig. 6C).

Based on these observations, we downregulated *Etv4* or *Etv5* expression using two independent shRNAs in KP as well as KPCic cell lines. As depicted in Appendix Fig. S3, individual downregulation of either gene had no effect on colony formation in the absence of trametinib and only moderately affected colony growth in its presence, albeit with slightly stronger growth inhibition after knockdown of *Etv5*. Hence, we assessed the consequences of the combined knockdown of *Etv4* and *Etv5*. While their combined absence faintly reduced colony formation in KP cell lines, it strongly inhibited colony growth in KPCic cell lines and re-sensitized CIC-deficient cells to treatment with trametinib (Fig. 6D–F). Likewise, overexpression of ETV4 also promoted resistance to trametinib (Fig. EV5E,F). Interestingly, knockdown of either *ETV4* or *ETV5* strongly inhibited the growth of human *KRAS*-mutant lung cancer cell lines, suggesting that the requirement for these transcription factors extends to human lung cancer (Fig. 6G,H). Together, these results indicate that ETV4 and ETV5 are involved in tumor cell proliferation and are the main mediators of drug resistance upon inactivation of CIC.

## Genetic inactivation of CIC creates selective vulnerabilities to overcome resistance

Since CIC inactivation promotes resistance to MAPK pathway inhibitors, we reasoned that it may also expose specific vulnerabilities that could be exploited to selectively block the proliferation of tumor cells that have lost CIC activity. To this end, we used a collection of 114 cancer drugs to identify compounds that selectively interfere with the proliferation of KPCic cells (Dataset EV4). We identified three drugs that significantly inhibited the growth of KPCic cells by more than 20% when compared to KP cell lines (Fig. 7A). These drugs (quizartinib, Tx-1123 and PFK15) were then validated individually in dose-response experiments in three independent KP and KPCic cell lines (Fig. 7B). While quizartinib did not exhibit substantial activity in KP or KPCic cell lines, Tx-1123 and PFK15 more strongly affected the survival of KPCic cells (Fig. 7B). Given their selectivity for KPCic cell lines, we next explored whether Tx-1123 and PFK15 were able to revert resistance to trametinib due to genetic CIC inactivation. As depicted in Fig. 7C, treatment with Tx-1123 and PFK15 had little effect on the viability of KP cell lines upon treatment with trametinib, especially at high doses. However, they strongly enhanced the response to

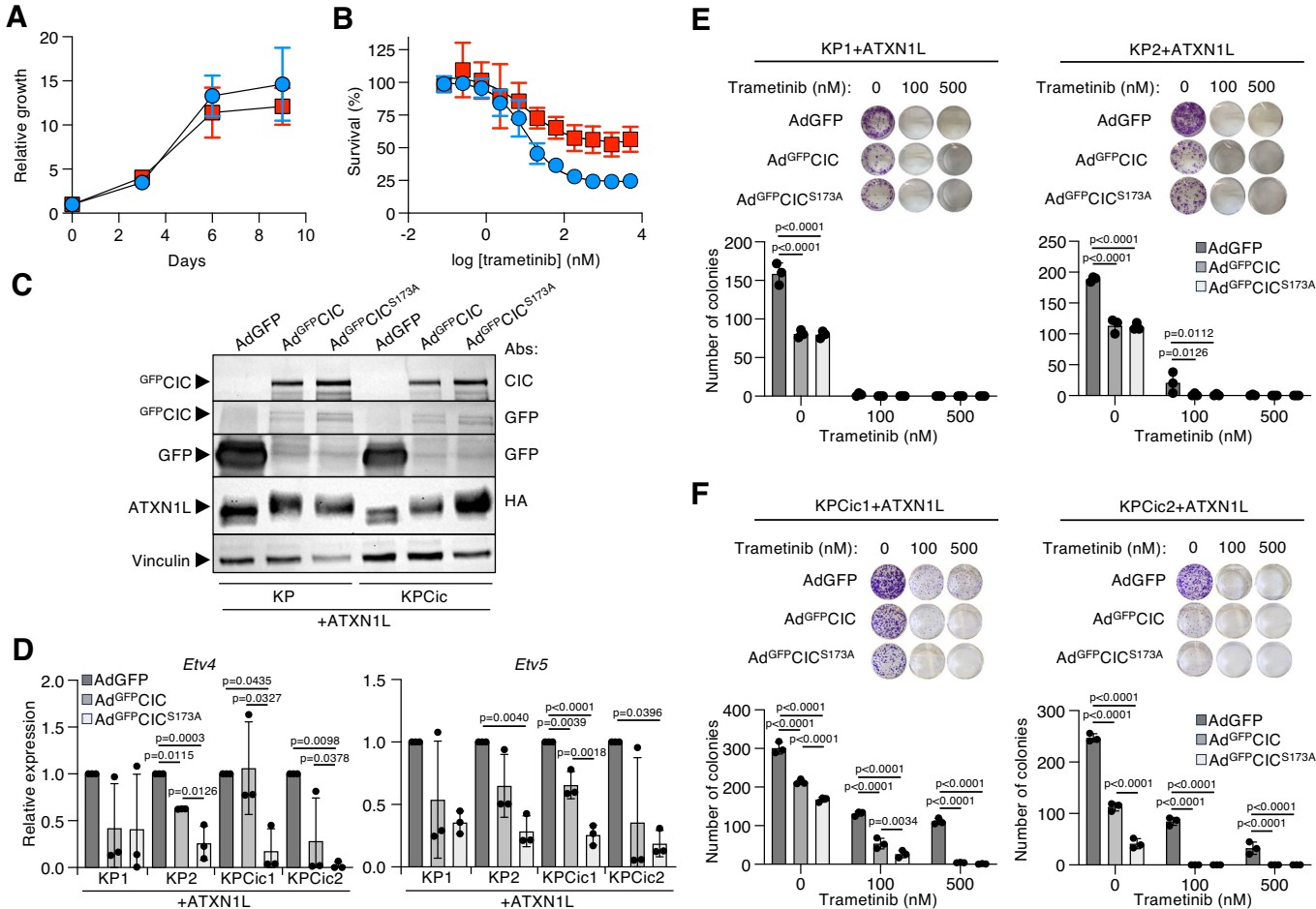

**Figure 5. CIC reactivation reduces tumor cell proliferation and reverts resistance to MEK inhibitors.**

(A) Proliferation of KP (blue circles, $n = 3$) and KPCic (red squares, $n = 3$) tumor cell lines. Results are shown as mean ± SD. (B) Relative viability of KP (blue circles, $n = 3$) and KPCic (red squares, $n = 3$) tumor cell lines after treatment with the indicated concentrations of trametinib for 72 h. Results are shown as mean ± SD. (C) Western blot analysis of $^{GFP}$CIC and GFP expression in KP as well as KPCic cells stably expressing HA-ATXN1L 72 h after infection with Ad-GFP, Ad-$^{GFP}$CIC and Ad-$^{GFP}$CIC$^{S173A}$. Antibodies against CIC, GFP and HA were used. Vinculin expression served as a loading control. (D) qRT-PCR of *Etv4* and *Etv5* expression levels in two independent KP as well as KPCic cell lines stably expressing HA-ATXN1L 72 h after infection with Ad-GFP (dark gray bars), Ad-$^{GFP}$CIC (gray bars) or Ad-$^{GFP}$CIC$^{S173A}$ (light gray bars). Results are shown as mean ± SD. Statistics, one-way ANOVA with Tukey's multiple comparison test ($n = 3$ biological replicates). (E) Representative images and quantification of colony formation assays in two independent KP cell lines stably expressing HA-ATXN1L after infection with Ad-GFP (dark gray bars), Ad-$^{GFP}$CIC (gray bars) or Ad-$^{GFP}$CIC$^{S173A}$ (light gray bars). Cells were continuously treated with the indicated concentrations of trametinib. Results are shown as mean ± SD. Statistics, Two-way ANOVA with Tukey's multiple comparison test ($n = 3$ technical replicates). (F) Representative images and quantification of colony formation assays in two independent KPCic cell lines stably expressing HA-ATXN1L after infection with Ad-GFP (dark gray bars), Ad-$^{GFP}$CIC (gray bars) or Ad-$^{GFP}$CIC$^{S173A}$ (light gray bars). Cells were continuously treated with the indicated concentrations of trametinib. Results are shown as mean ± SD. Statistics, two-way ANOVA with Tukey's multiple comparison test ($n = 3$ technical replicates). Source data are available online for this figure.

trametinib in KPCic cell lines to a level observed in KP cells. Next, we aimed to recapitulate the effect of PFK15 in human lung cancer cells. To this end, we eliminated CIC from the PDX-derived cell line PDX-dc1 (Sanclemente et al, 2018) using CRISPR/Cas9. As illustrated in Fig. 7D,E, elimination of CIC triggered resistance to trametinib, which was reverted by the addition of PFK15. More importantly, PFK15 strongly reduced the survival of trametinib-resistant patient-derived organoids that expressed low CIC and high ETV5 levels, while trametinib-resistant organoids showing high CIC and low ETV5 expression levels were not affected (Fig. 7F,G; Appendix Fig. S4). Together, these results indicate that genetic inactivation or absence of CIC creates vulnerabilities that can be exploited pharmacologically.

## Discussion

Activation of the MAPK pathway is critical for KRAS-driven LUAD development (Drosten and Barbacid, 2020). However, it has remained unknown how the MAPK pathway contributes to tumor growth, especially downstream of ERK kinases. Our findings unveil a link between *KRAS* copy number gains, amplification of MAPK signaling and CIC inactivation during the initiation of KRAS-driven lung tumors, which also has significant therapeutic implications.

It was previously shown that KRAS-driven LUADs show stage-specific amplification of MAPK signaling, but the molecular basis for amplified signaling or its consequences were unclear (Chung

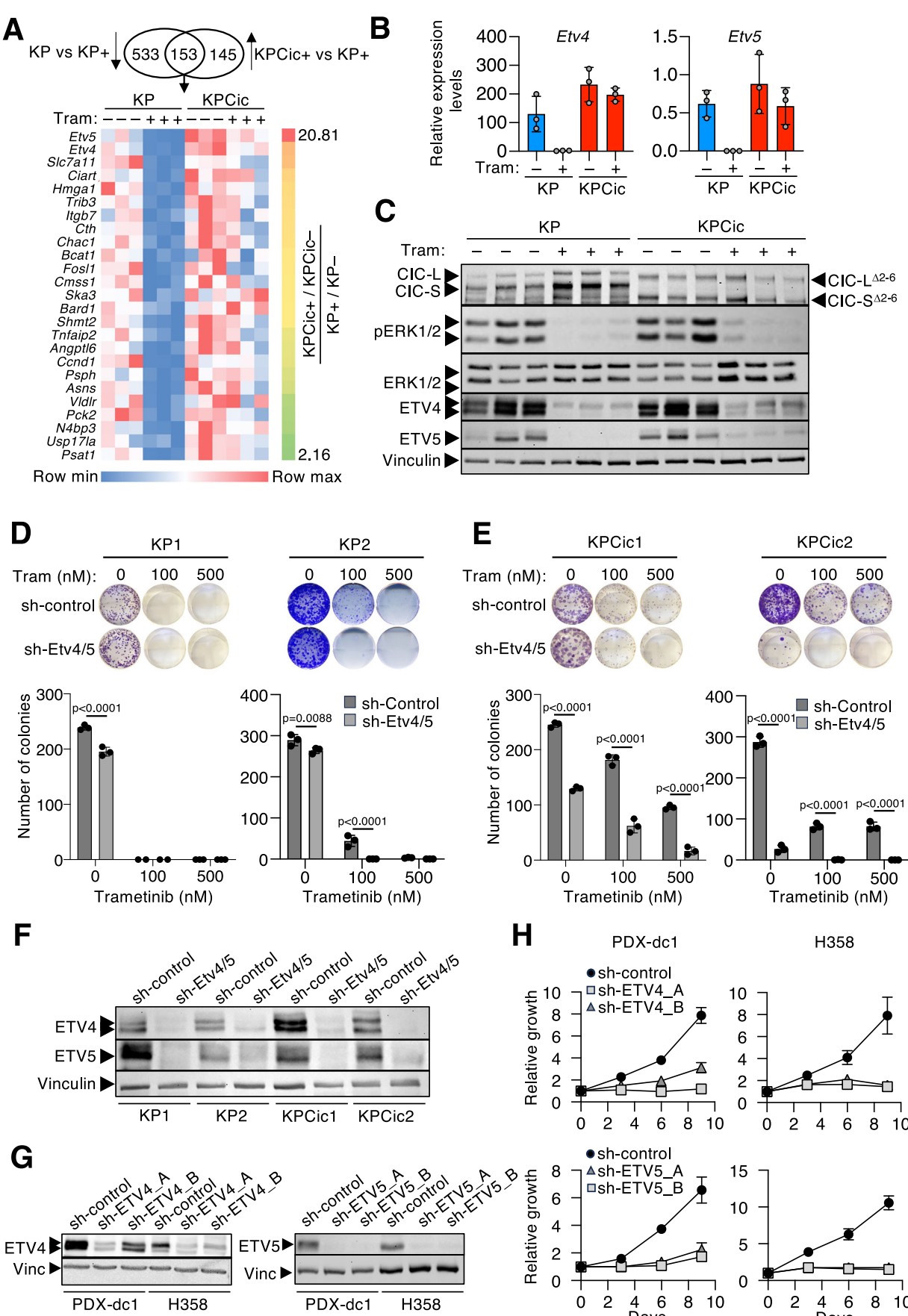

**Figure 6. Inhibition of ETV4 and ETV5 reduces proliferation and reverts resistance to MEK inhibitors.**

(A) Heatmap of gene expression in KP ($n = 3$) and KPCic ($n = 3$) cell lines treated with DMSO (−) or 20 nM trametinib (Tram, +) for 24 h. Genes were ranked according to a score calculated as (KPCic+/KPCic−)/(KP+/KP−) (right bar, from 2.16 to 20.81). (B) qRT-PCR of *Etv4* and *Etv5* expression levels in KP (blue bars, $n = 3$) and KPCic cell lines (red bars, $n = 3$) treated with DMSO (−) or 20 nM trametinib (Tram, +) for 24 h. Results are shown as mean ± SD. (C) Western blot analysis of CIC, pERK1/2, ERK1/2, ETV4, and ETV5 expression levels in KP ($n = 3$) and KPCic cell lines ($n = 3$) treated with DMSO (−) or 20 nM trametinib (Tram, +) for 24 h. Vinculin expression served as a loading control. (D) Representative images and quantification of colony formation assays in two independent KP cell lines stably expressing a control shRNA (dark gray bars) or a combination of shRNAs targeting *Etv4* and *Etv5* (light gray bars). Cells were continuously treated with the indicated concentrations of trametinib. Results are shown as mean ± SD. Statistics, Two-way ANOVA with Sidak's multiple comparison test ($n = 3$ technical replicates). (E) Representative images and quantification of colony formation assays in two independent KPCic cell lines stably expressing a control shRNA or a combination of shRNAs targeting *Etv4* and *Etv5*. Cells were continuously treated with the indicated concentrations of trametinib. Results are shown as mean ± SD. Statistics, two-way ANOVA with Sidak's multiple comparison test ($n = 3$ technical replicates). (F) Western blot analysis of ETV4 and ETV5 expression in 2 independent KP as well as KPCic cell lines stably expressing a control shRNA or a combination of shRNAs targeting *Etv4* and *Etv5*. Vinculin expression served as a loading control. (G) Western blot analysis of ETV4 and ETV5 expression in human PDX-dc1 and H358 cells stably expressing a control shRNA, two independent shRNAs targeting *ETV4* (left) or two independent shRNAs targeting *ETV5* (right). (H) Proliferation of PDX-dc1 and H358 cells stably expressing a control shRNA, two independent shRNAs targeting *ETV4* or two independent shRNAs targeting *ETV5* ($n = 3$ technical replicates). Results are shown as mean ± SD. Source data are available online for this figure.

et al, 2017; Cicchini et al, 2017; Chen et al, 2019; Feldser et al, 2010). Here, we show that progressive allelic imbalance of mutant *Kras* correlates with amplified MAPK signaling in *Kras/Trp53* mutant mice. Allelic imbalance, which can develop either through copy number gains of mutant *Kras* or LOH of the wild-type allele, is known to enhance tumor initiation in various contexts (Ambrogio et al, 2018; Najumudeen et al, 2024; Zhang et al, 2001). Our results also offer a plausible explanation for *Kras* copy number gains and amplified MAPK signaling. Since concomitant genetic inactivation of *Cic* abrogates *Kras* allelic imbalance and amplified MAPK signaling, we propose that *Kras* copy number gains may be selected for in tumors to promote stronger functional inactivation of CIC. In this model, *Kras* allelic imbalance leads to MAPK signal amplification and correspondingly stronger down-regulation of CIC, i.e., a graded signal-response mechanism that appears to be conserved from *Drosophila* to mammals (Jiménez et al, 2012; Rodríguez-Muñoz et al, 2022). Conversely, when oncogenic KRAS signaling is accompanied by genetic inactivation of CIC, the complete lack of CIC-mediated repression would allow tumor initiation without *KRAS* allelic imbalances (Appendix Fig. S5). Yet, suppression of *Kras* amplification in the absence of *Cic* is less evident in tumors retaining p53. The reason for this observation is currently unknown, but could be related with overall slower tumor progression rates in this model and the advanced age of the animals analyzed. Interestingly, when we inactivated *Cic* and *Trp53* alone or in combination in mice, tumors only formed when cells concomitantly lacking *Cic* and *Trp53* acquired latent *Kras* mutations, suggesting that the absence of p53 is required to expose the effect of *Cic* inactivation on tumor initiation.

In contrast to an earlier report (Okimoto et al, 2017), we found no evidence for an elevated metastatic potential of *Cic*-deficient tumor cells. The reasons for this discrepancy are also unclear, but our results are consistent with a more recent study demonstrating a role for CIC specifically during initiation of KRAS-driven lung tumors (Cai et al, 2021). Nevertheless, as opposed to the deletion of *Nf1* (Wang et al, 2019), for instance, the disruption of *Cic* only moderately enhances tumor initiation in our model. While loss of *Nf1* accelerates tumorigenesis via multiple mechanisms, including amplification of KRAS signaling and activation of FAK1, disruption of *Cic* will only affect a small subset of genes controlled by KRAS. Moreover, since the transcriptional profile of tumors lacking *Cic* obtained 5 months after infection does not substantially differ from those retaining *Cic*, notable derepression of CIC target genes might

be confined to a narrow window during the early steps of tumor initiation in which *Kras* amplifications do not yet occur.

Interestingly, functional CIC inactivation may not be the only critical consequence of amplified MAPK signaling, since concomitant elimination of the tumor suppressor *Rb1* also abrogates the requirement for amplified MAPK signaling in KRAS-driven LUAD (Walter et al, 2019). Interestingly, both CIC and RB1 proteins function as tumor suppressors that can be inactivated through phosphorylation, suggesting that amplified MAPK signaling might ensure maximal inactivation of CIC and RB1, together with other potential molecular events that contribute to the transformed phenotype. In this context, loss of CIC or RB1 alone may be sufficient to induce gene expression changes that facilitate tumor growth even without MAPK signaling amplification, especially at early stages of tumorigenesis.

Several studies have demonstrated that amplified MAPK signaling can promote transformation of Club cells, suggesting that these cells require a higher threshold of MAPK pathway activity for transformation (Cicchini et al, 2017; Nieto et al, 2017). Our data show that genetic inactivation of CIC phenocopies the effect of amplified MAPK signaling in Club cells, providing further evidence for CIC as a major target of *Kras* allelic imbalance and amplified MAPK signaling. Furthermore, our results align with a recently observed transient Club/AT2-marker double-positive cellular state that originates during lineage conversion upon oncogenic insult (Chen et al, 2022). A similar cellular state emerges during lung regeneration upon tissue damage, thus raising the possibility that CIC inactivation also contributes to lineage conversion of Club cells in this context (Zheng et al, 2013). Alternatively, this transient double-positive condition may reflect a common response to various, unrelated stress stimuli such as tissue damage or oncogenic signaling, including CIC inactivation.

*CIC* mutations have been identified in experimental models and patients that developed resistance to drugs targeting the MAPK pathway (Simón-Carrasco et al, 2017; Wang et al, 2017; Liao et al, 2017; Da Vià et al, 2020; Hashiba et al, 2020). In line with these observations, our results show that KP tumor cells expressing inactive CIC$^{\Delta 2\text{-}6}$ proteins are resistant to the MEK inhibitor trametinib. Interestingly, reconstitution of CIC repressor activity via ectopic expression of CIC$^{S173A}$, a mutant CIC protein that is less responsive to ERK-mediated inactivation, and ATXN1L, a co-repressor required for CIC stability and DNA binding (Lee et al, 2011; Wong et al, 2019), not only reduced tumor cell proliferation, but also restored sensitivity to trametinib. Concomitant ATXN1L

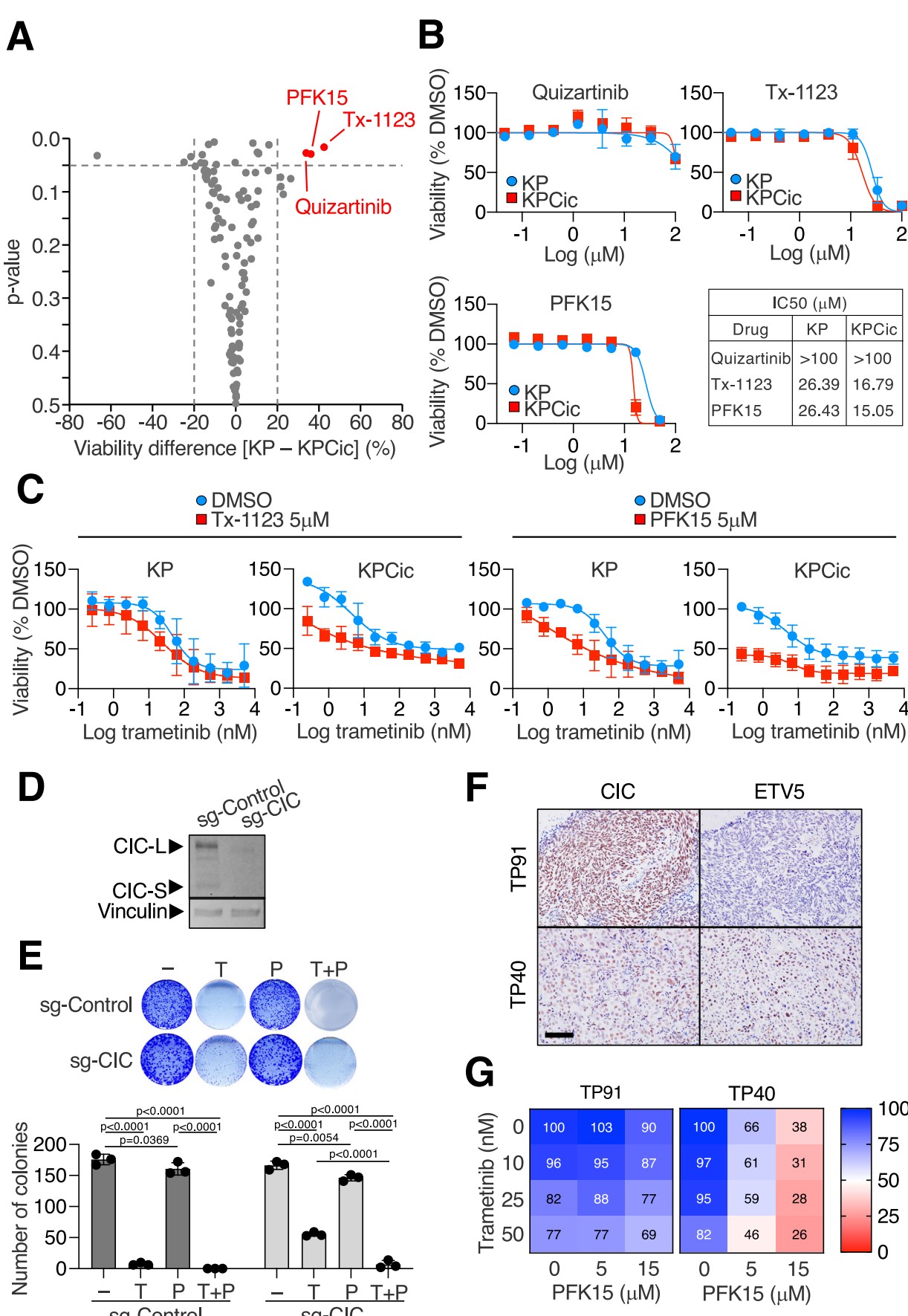

**Figure 7. Identification of drugs that preferentially inhibit tumor cells devoid of CIC activity.**

(A) Viability of KP (n = 3) and KPCic (n = 4) cell lines treated with a collection of compounds. Each compound was used at 5 μM for 72 h. Results are represented as mean viability differences. Statistics, unpaired t-test. (B) Relative viability and summary of IC50 values of KP (blue circles, n = 3) and KPCic (red squares, n = 3) tumor cell lines after treatment with the indicated concentrations of quizartinib, Tx-1123 and PFK15 for 72 h. Results are shown as mean ± SD. (C) Left, relative viability of KP (n = 2 cell lines for KP cells treated with Tx-1123; all others n = 3) and KPCic (n = 3) tumor cell lines after treatment with the indicated concentrations of trametinib in combination with 5 μM Tx-1123 (red squares) or DMSO (blue circles) for 72 h. Right, relative viability of KP (n = 3) and KPCic (n = 3) tumor cell lines after treatment with the indicated concentrations of trametinib in combination with 5 μM PFK15 (red squares) or DMSO (blue circles) for 72 h. Results are shown as mean ± SD. (D) Western blot analysis of CIC expression in PDX-dc1 cells infected with lentiviral vectors expressing Cas9 and a control sgRNA or an sgRNA targeting CIC. Vinculin expression served as a loading control. (E) Representative images and quantification of colony formation assays in PDX-dc1 cells infected with lentiviral vectors expressing Cas9 and a control sgRNA (dark gray bars) or an sgRNA targeting CIC (light gray bars), either treated with DMSO (−), 100 nM trametinib (T), 1 μM PFK15 (P), or 100 nM trametinib + 1 μM PFK15 (T + P). Results are shown as mean ± SD. Statistics, two-way ANOVA with Sidak's multiple comparison test (n = 3 technical replicates). (F) Representative images of CIC and ETV5 immunostaining in sections from PDX tumors. Scale bar, 100 μm. (G) Heatmap indicating viability of patient-derived organoids (% DMSO) treated with the indicated concentration of trametinib and PFK15 (n = 3 biological replicates). Source data are available online for this figure.

overexpression was only required upon adenoviral expression of CIC variants, since endogenous *Atxn1/Atxn1l* levels, albeit reduced, were sufficient to sustain CIC's repressor activity in KP cells. Overall, these observations demonstrate that repression of CIC target genes is required for the anti-tumor effects of MAPK pathway inhibitors. Indeed, the well-known CIC targets ETV4 and ETV5 are key mediators of tumor cell proliferation and drug resistance. Both *ETV4* and *ETV5* are frequently overexpressed in human NSCLC and are associated with poor patient prognosis (Wang et al, 2020; Cheng et al, 2019; Li et al, 2023). While both genes encode transcriptional activators, how exactly they contribute to these activities remains to be determined.

In addition, we reasoned that genetic inactivation or mutation of *Cic* may also expose specific vulnerabilities that can be exploited therapeutically. Indeed, we identified two drugs, Tx-1123 and PFK15, with preferential activity in *Cic*-deficient tumor cells, suggesting that these drugs may be an alternative for patients who develop resistance via mutation of *CIC*. While the pleiotropic effects of Tx-1123 make it difficult to predict how this drug could preferentially affect *Cic*-deficient cells (Hori et al, 2002), PFK15 specifically targets the rate-limiting glycolytic enzyme PFKFB3 (Shi et al, 2017; Clem et al, 2013). This suggests that *CIC*-deficient tumor cells are particularly dependent on glucose metabolism, likely because of constitutive derepression of specific target genes. Indeed, our data (Datasets EV1, EV3) as well as previous observations indicate that CIC controls multiple genes implicated in metabolic processes, suggesting that CIC-deficient cells could be particularly vulnerable to interference with metabolic requirements (Park et al, 2023; Wong et al, 2019).

## Methods

### Reagents and tools table

| Reagent/resource | Reference or source | Identifier or catalog number |
|---|---|---|
| **Experimental models** | | |
| *Kras*[+/LSLG12Vgeo] | Guerra et al, 2003 | N/A |
| *p53*[lox/lox] | Jonkers et al, 2001 | N/A |
| *Cic*[lox/lox] | Simón-Carrasco et al, 2017 | N/A |
| PDX-dc1 (*H. sapiens*) | Sanclemente et al, 2018 | N/A |
| NCI-H358 (*H. sapiens*) | ATCC | CRL-5807 |

| Reagent/resource | Reference or source | Identifier or catalog number |
|---|---|---|
| HEK293T (*H. sapiens*) | ATCC | CRL-3216 |
| 293A (*H. sapiens*) | Invitrogen | R70507 |
| PDX TP40 (*H. sapiens*) | Garmendia et al, 2019 | N/A |
| PDX TP91 (*H. sapiens*) | Quintanal-Villalonga et al, 2020 | N/A |
| KP cell lines 1–3 (*M. musculus*) | This study | *Kras*[+/G12V];*Trp53*[-/-] |
| KPCic cell lines 1–3 (*M. musculus*) | This study | *Kras*[+/G12V];*Trp53*[-/-];*Cic*[D2-6/D2-6] |
| **Recombinant DNA** | | |
| Ad5CMVCre | University of Iowa Viral Vector Core | VVC-U of Iowa-5 |
| Ad5CC10-Cre | University of Iowa Viral Vector Core | VVC-Berns-1166 |
| Ad-GFP | Q-BIOgene | |
| Ad-GFP-CIC | This study | |
| Ad-GFP-CIC[S173A] | This study | |
| pLKO.1-puro sh-Etv4_1890 (*M. musculus*) | Sigma-Aldrich | TRCN0000295466 |
| pLKO.1-puro sh-Etv4_1137 (*M. musculus*) | Sigma-Aldrich | TRCN0000306806 |
| pLKO.1-puro sh-Etv5_1762 (*M. musculus*) | Sigma-Aldrich | TRCN0000312867 |
| pLKO.1-puro sh-Etv5_1646 (*M. musculus*) | Sigma-Aldrich | TRCN0000312868 |
| pLKO.1-blast sh-Etv4_1137 (*M. musculus*) | This study | N/A |
| pLKO.1-puro sh-ETV4_A (*H. sapiens*) | Sigma-Aldrich | TRCN000013937 |
| pLKO.1-puro sh-ETV4_B (*H. sapiens*) | Sigma-Aldrich | TRCN000013934 |
| pLKO.1-puro sh-ETV5_A (*H. sapiens*) | Sigma-Aldrich | TRCN000013938 |
| pLKO.1-puro sh-ETV5_B (*H. sapiens*) | Sigma-Aldrich | TRCN000013930 |
| non-targeting shRNA vector | Sigma-Aldrich | SHC002 |
| lentiCRISPRv2-sgCIC#1 | Simón-Carrasco et al, 2017 | N/A |

| Reagent/ resource | Reference or source | Identifier or catalog number |
|---|---|---|
| pVLX-puro HA-ATXN1L | This study | N/A |
| pLPC-ETV4-HA | This study | N/A |
| RP24-359O2 | BACPAC Resource Centre | N/A |
| RP24-175B6 | BACPAC Resource Centre | N/A |
| RP24-345H19 | BACPAC Resource Centre | N/A |
| RP24-275J18 | BACPAC Resource Centre | N/A |
| **Antibodies** | | |
| Ki67 | Cell Signaling | 12202 |
| pERK1/2 | Cell Signaling | 9101 |
| ERK1/2 | Cell Signaling | 4695 |
| SPC | Cell Signaling | 9661 |
| CIC | Invitrogen | PA5-83721 |
| ETV4 | Proteintech | 10684-1-AP |
| ETV5 | Proteintech | 13011-1-AP |
| ETV5 | Abcam | ab102010 |
| CC10 | Santa Cruz Biotechnology | sc-0772 |
| TTF-1 | Epitomics | 2044-1 |
| GFP | Proteintech | 50430-AP |
| HA-Tag | GenScript | A01244 |
| GAPDH | Sigma-Aldrich | G8795 |
| pAKT | Cell Signaling | 9271 |
| AKT | Cell Signaling | 9272 |
| VINCULIN | Sigma-Aldrich | V9131 |
| LAMIN A/C | Santa Cruz Biotechnology | sc-376248 |
| MEK1 | Santa Cruz Biotechnology | sc-6250 |
| **Oligonucleotides and other sequence-based reagents** | | |
| Etv4_qPCR-F | CGACTCAGATGTCCCTGGAT | |
| Etv4_qPCR-R | GCCTGTCCAAGCAATGAAAT | |
| Etv5_qPCR-F | GGGAAATCTCGATCAGAGGAC | |
| Etv5_qPCR-R | GGAGCATGAAGCACCAAGTT | |
| Ccnd1_qPCR-F | TGCGTGCAGAAGGAGATTGT | |
| Ccnd1_qPCR-R | CTCACAGACCTCCAGCATCCA | |
| Fosl1_qPCR-F | AAACCGAAGAAAGGAGCTGACA | |
| Fosl1_qPCR-R | CCCGATTTCTCATCCTCCAA | |
| Spry2_qPCR-F | CAGAGTTTGGAAAGAAGGAAAAAGTT | |
| Spry2_qPCR-R | CAGGTCTTGGCAGTGTGTTCA | |
| Spry4_qPCR-F | TGGAGCGATGCTTGTGACTCT | |
| Spry4_qPCR-R | GAGGCTGGAGGTCCTGAACTG | |
| Dusp6_qPCR-F | CGGATCACTGGAGCCAAAAG | |
| Dusp6_qPCR-R | GCCTCGGGCTTCATCTATGA | |
| Myc_qPCR-F | ACTCACCAGCACAACTACGC | |
| Myc_qPCR-R | TTGCTGATCTGCTTCAGGAC | |
| Atxn1_qPCR-F | CTTCTACGCTGGCACTCAAC | |
| Atxn1_qPCR-F | CAGACCACCAGCATAGGTGA | |
| Atxn1l_qPCR-R | AGTCACTTCTGATGCCCGAA | |
| Atxn1l_qPCR-R | ATATCCTCGCTGGTCACAGG | |
| b-Actin_qPCR-F | GACGGCCAGGTCATCACTATTG | |
| b-Actin_qPCR-R | AGGAAGGCTGGAAAAGAGCC | |
| KrasG12Vlox-F | GGAACTTCGCATGCATAACTTCGTATAATGT | |
| KrasG12Vlox-R | CAAAGCACGGATGGCATCTTGGACC | |
| KrasWT/G12V-F | GATACATTCCTTTGAGAGCCATT | |
| KrasWT/G12V-R | CTCAGTCATTTTCAGCAGGC | |

| Reagent/ resource | Reference or source | Identifier or catalog number |
|---|---|---|
| Etv4_ChIP-F | GCTTCCGTCTTTTTTTCTCAATT | |
| Etv4_ChIP-R | ATCAGAAAGTAGGGGCTGGC | |
| Etv5_ChIP-F | TTGCTCCTGATCACACATGC | |
| Etv5_ChIP-R | GCTGGAACCTCGTGAATGAT | |
| Ccnd1_ChIP-F | AAATTTGCATGAGCCAATCC | |
| Ccnd1_ChIP-R | GCAGAGCTCAACGAAGTTCC | |
| Cdk1_ChIP-F | GGCCTTCAACGTATGAATTAGC | |
| Cdk1_ChIP-R | AGTTGGTATTGCACATAAGTCT | |
| **Chemicals, Enzymes and other reagents** | | |
| Trametinib | MedChemExpress | HY-10999 |
| PFK15 | Sigma-Aldrich | SML1009 |
| TX-1123 | Sigma-Aldrich | 655200 |
| RNeasy Mini Kit | Qiagen | 74104 |
| PureLink RNA Mini Kit | Invitrogen | 12183020 |
| NZY First-Strand cDNA Synthesis kit | NZYTech | MB12501 |
| PowerTrack SYBR Green PCR Master Mix | Applied Biosystems | 16525231 |
| Cell Titer Glo Luminescent Cell Viability Kit | Promega | G7570 |
| QuikChange II Site-Directed Mutagenesis Kit | Agilent | 200523 |
| DreamTaq Green PCR Master Mix | Thermo Fisher Scientific | K1081 |
| NE-PER Nuclear and Cytoplasmic Extraction Reagents | Thermo Fisher Scientific | 78833 |
| Histology FISH Accessory Kit | Agilent (DAKO) | K5799 |
| TOPO TA-Cloning Kit | Invitrogen | 450030 |
| Protein A/G PLUS agarose | Santa Cruz Biotechnology | sc-2003 |
| AflII | Thermo Fisher Scientific | ER0831 |
| NotI | Thermo Fisher Scientific | ER0592 |
| PmeI | New England Biolabs | R0560S |
| EcoRI | Thermo Fisher Scientific | ER0721 |
| Collagenase | Sigma-Aldrich | C9891 |
| DNase | Sigma-Aldrich | D5025 |
| Dispase | Life Technologies | 17105041 |
| Advanced DMEM/F12 | Gibco | 12634010 |
| HEPES | Life Technologies | 1560106 |
| Glutamine | Sigma-Aldrich | G8541 |
| Matrigel | Corning | CLS356231 |
| EGF | Life Technologies | PMG8041 |
| Insulin | Sigma-Aldrich | I3536 |
| Hydrocortisone | Sigma-Aldrich | H0888 |
| B27 | Life Technologies | 17504-044 |
| Y-27632 | Sigma-Aldrich | SCM075 |
| Formaldehyde Pierce™16% (p/v) | Thermo Fisher Scientific | 28908 |
| Glycine | Sigma-Aldrich | G7126 |

| Reagent/resource | Reference or source | Identifier or catalog number |
|---|---|---|
| cOmplete Protease Inhibitor | Sigma-Aldrich | 11836153001 |
| HEPES-KOH | Sigma-Aldrich | H3375-100G |
| LiCl | Sigma-Aldrich | L4408 |
| NP-40 | Sigma-Aldrich | 492016-500 |
| Triton X-100 | Sigma-Aldrich | T9284 |
| N-lauroyl sarcosine | Sigma-Aldrich | L5125-50G |
| Sodium deoxycholate | Sigma-Aldrich | D6750 |
| Dynabeads Protein A | Thermo Fisher Scientific | 10001D |
| RNase A | Sigma-Aldrich | 10109142001 |
| Proteinase K | Roche | 03115879001 |
| Phenol-chloroform | Sigma-Aldrich | 77617 |
| DMEM (1X) | Gibco | 41966-029 |
| RPMI Medium 1640 (1X) | Gibco | 21875-034 |
| **Software** | | |
| QuPath 0.4.2 | https://qupath.github.io | |
| GraphPad Prism 8.4.0 | GraphPad Software, Inc | |
| BBtools | BBMap, https://sourceforge.net/projects/bbmap/ | |
| STAR v2.5 | Dobin et al, 2013 | |
| HTSeq | Anders et al, 2015 | |
| DESeq2 | Love et al, 2014 | |
| BWA-MEM 0.7.15 | github.com/lh3/bwa.git | |
| Picard tools 2.9.0 | broadinstitute.github.io/picard | |
| Integrative Genomics Viewer | igv.org | |
| MORPHEUS | https://software.broadinstitute.org/morpheus, Broad Institute | |
| ImageJ | https://imagej.nih.gov/ij/index.html | |
| StringTie | ccb.jhu.edu/software/stringtie | |
| **Other** | | |
| Bioruptor Plus | Diagenode | |
| Qubit 4 Fluorometer | Invitrogen | |
| QuantStudio™ 3-96-well 0.1 mL Block | Thermo Fisher Scientific | QS3_272312003 |

## Mouse strains and tumor induction

$Kras^{\text{LSLG12Vgeo}}$ (Guerra et al, 2003), $Trp53^{\text{lox}}$ (Jonkers et al, 2001), and $Cic^{\text{lox}}$ (Simón-Carrasco et al, 2017) alleles have been published. To induce lung tumor formation, 8–12 weeks old mice were intranasally infected as described (Esteban-Burgos et al, 2020) with $5 \times 10^5$ or $5 \times 10^7$ pfu Ad-Cre or $5 \times 10^7$ pfu Ad-CC10-Cre (Sutherland et al, 2011). Adenoviral vectors were purchased from the Viral Vector Core at the University of Iowa. Animals were maintained in a mixed 129/Sv-C57BL/6 background. Female and male mice were used for the experiments, and no blinding was

done. No sample size estimations were performed. All animal experiments were approved by the Ethical Committees of the Spanish National Cancer Research Centre (CNIO), the Carlos III Health Institute and the Autonomous Community of Madrid (PROEX 161/14), or the Bioethics Committee of the University of Salamanca as well as the Castilla y Leon Autonomous Government (716 CSIC-USAL) and were performed in accordance with the guidelines stated in the International Guiding Principles for Biomedical Research Involving Animals (CIOMS). Mice were housed in specific-pathogen-free conditions at the Animal Facilities of the CNIO (Association for Assessment and Accreditation of Laboratory Animal Care, JRS: dpR 001659) or the Cancer Research Center (CIC) in Salamanca.

## Cell lines and treatments

Tumor cell lines were established from tumors in KP or KPCic mice 5 months after infection with Ad-Cre and grown in DMEM supplemented with 5% FBS. The human PDX-derived cell line PDX-dc1 was previously described (Sanclemente et al, 2018) and maintained in DMEM supplemented with 10% FBS. NCI-H358 cells were obtained from the ATCC, grown in RPMI-1640 medium supplemented with 10% FBS, and authenticated using the CLA IdentiFiler Plus kit (Thermo Fisher Scientific). All cell lines were routinely tested for mycoplasma contamination. Trametinib was purchased from MedChemExpress and used at the indicated concentrations. PFK15 and Tx-1123 were purchased from Sigma-Aldrich.

## Histopathology and immunohistochemistry

Tissues were fixed in 10%-buffered formalin (Sigma-Aldrich) and embedded in paraffin. For histopathological visualization, 2.5 µm tissue sections were stained with Hematoxylin & Eosin (H&E) and tumors were classified according to standard histopathological grade criteria (Jackson et al, 2005). Antibodies used for immunostaining included those raised against: Ki67 (Cell Signaling Technology, 12202, 1:50), phospho-ERK1/2 (Cell Signaling Technology, 9101, 1:300), SPC (Abcam, ab212326, 1:4000), CIC (Invitrogen, PA5-83721, 1:100), ETV5 (Proteintech, 13011-1-AP, 1:300), CC10 (Santa Cruz Biotechnology, sc-9772, 1:1000) and TTF-1 (Epitomics, 2044-1, 1:100). For imaging analysis, slides were scanned on a ZEISS Axio Scan.Z1 scanner and processed using QuPath version 0.4.2 software.

## Western blot analysis

Proteins were extracted in protein lysis buffer (50 mM Tris-HCl pH 7.5, 150 mM NaCl, 1% NP-40) supplemented with cOmplete Mini protease inhibitors (Roche). A total of 30 µg protein extracts were separated by SDS-PAGE, transferred to a nitrocellulose blotting membrane (GE Healthcare) and blotted with antibodies raised against: CIC (Invitrogen, PA5-83721, 1:1000), ETV4 (Proteintech, 10684-1-AP, 1:1000), ETV5 (Abcam, ab102010, 1:1000), ERK1/2 (Cell Signaling Technology, 4695, 1:1000), phospho-ERK1/2 (Cell Signaling Technology, 9101, 1:500), AKT (Cell Signaling Technology, 9272, 1:1000), phospho-AKT (Cell Signaling Technology, 9271, 1:1000), GFP-tag (Proteintech, 50430-AP, 1:1000), HA-tag (GenScript, A01244, 1:1000), GAPDH (Sigma-Aldrich, G8795, 1:5000) and Vinculin (Sigma-Aldrich, V9131, 1:5000).

## Subcellular fractionation

Nuclear and cytoplasmic fractions were extracted with the NE-PER Nuclear and Cytoplasmic Extraction Reagents (Thermo Fisher Scientific) following the manufacturer's guidelines. Efficient nuclear and cytoplasmic fractionation was confirmed by Western blot analysis using Lamin A/C antibodies (Santa Cruz Biotechnology, sc-376248, 1:1000) for the nuclear fraction and MEK1 antibodies (Santa Cruz Biotechnology, sc-6250, 1:500) for the cytoplasmic fraction.

## Knockdown and CRISPR/Cas9 assays

Tumor cells were infected with lentiviral supernatants expressing shRNA against mouse *Etv4* (TRCN0000295466 [sh-Etv4_1890] and TRCN0000306806 [sh-Etv4_1137], Sigma-Aldrich) and mouse *Etv5* (TRCN0000312867 [sh-Etv5_1762] and TRCN0000312868 [sh-Etv5_1646], Sigma-Aldrich). For combined knockdowns, sh-Etv4_1137 was cloned into a pLKO.1 plasmid that carries a blasticidin resistance cassette and used in combination with sh-Etv5_1762. For knockdown experiments in human cell lines, cells were infected with lentiviral supernatants expressing shRNA against human *ETV4* (TRCN000013937 [sh-ETV4_A] and TRCN000013934 [sh-ETV4_B], Sigma-Aldrich) or human *ETV5* (TRCN000013938 [sh-ETV5_A] and TRCN000013930 [sh-ETV5_B], Sigma-Aldrich). A non-targeting shRNA vector (SHC002) was used as a negative control. A lentiviral vector expressing an sgRNA targeting human *CIC* (lentiCRISPRv2-sgCIC#1) has been described previously (Simón-Carrasco et al, 2017).

## Construction of viral vectors expressing GFP-CIC variants, HA-ATXN1L, and ETV4-HA

Adenoviral vectors expressing $^{GFP}$CIC or $^{GFP}$CIC$^{S173A}$ were generated using the AdEasy system (He et al, 1998). In brief, the $^{GFP}$CIC cDNA was excised from pcDNA5/FRT/TO-GFP-CIC (Dissanayake et al, 2011) by digesting the DNA with AflII and NotI. An AflII restriction site also was introduced 5' of the NotI restriction site in pShuttle-CMV (He et al, 1998) by site-directed mutagenesis, and the $^{GFP}$CIC cDNA was subsequently cloned into this modified pShuttle-CMV vector after digestion with AflII and NotI. The S173A mutation was then introduced by site-directed mutagenesis. Adenoviral vectors were finally retrieved by homologous recombination of pAdEasy1 with the respective pShuttle constructs in *Escherichia coli* BJ5183 and introduction of the PmeI-linearized pAdEasy1-$^{GFP}$CIC and pAdEasy1-$^{GFP}$CIC$^{S173A}$ into 293A cells. Adenoviral infections were carried out at multiplicities of infection (moi) of 50. Lentiviral vectors expressing HA-ATXN1L were generated by amplifying the human ATXN1L cDNA by PCR with forward primers including an EcoRI restriction site and an HA-tag following the ATG start codon and a reverse primer including an EcoRI site after the STOP codon. PCR products were cloned into pCR2.1 using the TOPO TA-Cloning Kit (Invitrogen) and sequence-verified. Finally, the HA-ATXN1L cDNA was excised with EcoRI and cloned into pLVXpuro (Clontech) after digestion with EcoRI. The correct orientation was confirmed by DNA sequencing. Retroviral vectors expressing ETV4-HA were generated by PCR-amplification of the human ETV4 cDNA from pCMVSport6 ETV4 (BC016623) with forward primers including an EcoRI restriction site and reverse primers adding an HA-tag preceding the STOP codon followed by an EcoRI restriction site. The resulting PCR product was digested with EcoRI and cloned into pLPC after digestion with EcoRI. The correct orientation was confirmed by DNA sequencing.

## qRT-PCR

Total RNA was extracted with the RNeasy Mini Kit (Qiagen) and reverse-transcribed using the NZY First-Strand cDNA Synthesis kit (NZYTech) following the manufacturer's instructions. Quantitative real-time PCR reactions were performed on a QuantStudio 6 Flex Real-Time PCR System (Applied Biosystems) using the Power-Track SYBR Green PCR Master Mix (Applied Biosystems). Values were quantified according to the $\Delta\Delta Ct$ method, and β-actin was used for normalization.

## PCR analysis to verify *Kras*$^{G12Vlox}$ excision

PCRs were performed on genomic DNA from K$^{G12Vlox}$PC2 cells (Salmón et al, 2023) using specific forward (5'-GGAACTTCG-CATGCATAACTTCGTATAATGT-3') and reverse primers (5'-CAAAGCACGGATGGCATCTTGGACC-3') for the *Kras*$^{G12Vlox}$ allele, yielding a 594 bp band corresponding to the unexcised allele and a 170 bp band for the excised allele. PCRs were performed using the DreamTaq Green PCR Master Mix (Thermo Scientific) with the following PCR conditions: 30 s 94 °C, 30 s 60 °C, 30 s 72 °C (35 cycles).

## PCR analysis of *Kras* allele abundance

The relative abundance of *Kras*$^{WT}$ and *Kras*$^{G12V}$ alleles was estimated by semiquantitative PCR with the following PCR conditions: 30 s 94 °C, 30 s 60 °C, 30 s 72 °C (20 cycles); forward primer (5'-GATACATTCCTTTGAGAGCCATT-3'), reverse primer (5'-CTCAGTCATTTTCAGCAGGC-3'). This PCR yielded DNA fragments of 309 bp for the *Kras*$^{WT}$ allele and 398 bp for the recombined *Kras*$^{G12V}$ allele (Guerra et al, 2003). PCR band intensities were quantified with ImageJ.

## Fluorescence-in situ hybridization (FISH) assays

Two sets of bacterial artificial chromosome (BAC) clones (RP24-359O2 and RP24-175B6 for *Kras* at 6qG3; and RP24-345H19 and RP24-275J18 for control probe at 6qA3.3) were obtained from the BACPAC Resources Centre (https://bacpacresources.org/) to generate a FISH probe to detect *Kras* amplification or chromosome 6 aneuploidies. The *Kras* probe was labeled with Spectrum-Orange and the control probe with Spectrum-Green, using the NICK translation assay (Abbot Molecular). FISH analyses were performed as previously described (Martinez-Lage et al, 2020) on 2.5 μm tissue sections mounted on positively charged slides (SuperFrost, Thermo Scientific). Briefly, the slides were first deparaffined in xylene and gradually rehydrated through a series of ethanol washes. The Histology FISH Accessory Kit (DAKO) was used following the manufacturer's guidelines. The process included pre-treatment in 2-[N-morpholino]ethanesulphonic acid (MES), followed by protein digestion in a pepsin solution. After dehydration, the samples were

denatured with the specific probe at 66 °C for 10 min and allowed to hybridize overnight at 45 °C in a DAKO hybridizer. The slides were then washed with a 20×SSC (saline-sodium citrate) buffer containing Tween-20 detergent at 63 °C and mounted in a fluorescence mounting medium (DAPI). FISH signals were manually counted within the nuclei throughout the tissue, and images were captured using a CCD camera (Photometrics SenSys camera) connected to a PC running the Zytovision image analysis system (Applied Imaging Ltd.) with focus motor and Z-stack software. The z-stack images were manually scored by two independent investigators by counting the number of co-localized signals.

## Compound screening

A collection of 114 drugs covering multiple oncogenic pathways (see Dataset EV4) was used at a concentration of 5 μM in KP and KPCic cell lines. 3000 cells were seeded in 96-well plates, and the compounds were added the following day. Cell viability was assayed after 72 h using the Cell Titer Glo Luminescent Cell Viability Kit (Promega, G7571). Luminescence counts were read in a Victor plate reader (Perkin Elmer).

## Generation of PDX-derived organoids

The lung PDX tumors TP40 and TP91 have been described previously (Garmendia et al, 2019; Quintanal-Villalonga et al, 2020). To generate organoids, PDX tumors were enzymatically digested with 1.2 mg/ml collagenase (Sigma-Aldrich, C9891), 10 mg/ml DNase (Sigma-Aldrich, D5025), and 0.125 mg/ml dispase (Life Technologies, 17105041) in Basic medium (Advanced DMEM/F12 [Gibco, 12634010], 1xHEPES [Life Technologies, 1560106], 1xGlutamine [Sigma-Aldrich, G8541]) at 37 °C in agitation for 1 h. After incubation, the digested tumor pieces were filtered using 70 mm filters, and the disaggregated cells were centrifuged at 1500 rpm for 5 min. After two washes with Basic medium, live cells were counted.

## Organoid treatments

Organoids were treated with a drug matrix using 0, 10, 25, and 50 nM trametinib and 0, 5, and 15 μM PFK15 in triplicate for 7 days. 5000 cells/well were resuspended in 36 μl Matrigel (Corning, CLS356231) and seeded in a 96-well plate. Cells were allowed to solidify at 37 °C in a $CO_2$ (5%) incubator for 10 min and 150 μl/well Complete medium (Basic Medium supplemented with 2% FBS, 3 ng/ml epidermal growth factor [Life Technologies, PMG8041], 5 mg/ml human insulin [Sigma-Aldrich, I3536], 1 mg/ml hydro-cortisone [Sigma-Aldrich, H0888], 1x B27 (Life Technologies, 17504-044), 10.5 μM ROCK inhibitor Y-27632 [Sigma-Aldrich, SCM075]) was added to each well. PFK15 and trametinib were added 24 h after seeding and refreshed at day 4. Cell viability was assayed after 7 days using the Cell Titer Glo Luminescent Cell Viability Kit (Promega, G7571). Luminescence counts were read in a Victor plate reader (Perkin Elmer).

## RNA sequencing and data analysis

Total RNA samples from cell lines were converted into sequencing libraries using the QuantSeq 3' mRNA-Seq Library Prep Kit (FWD) for Illumina (Lexogen, Cat. No. 015) and sequenced on an Illumina NextSeq 550 instrument. Sequencing read analysis was performed by an automated analysis service provided by Lexogen. Briefly, reads were successively processed with bbduk from BBTools (BBMap, https://sourceforge.net/projects/bbmap/), aligned to the GRCm38/mm10 genome assembly using STAR v2.5 (Dobin et al, 2013) and counted with HTSeq (Anders et al, 2015). Differential gene expression analysis was performed using DESeq2 (Love et al, 2014). For RNA sequencing of mouse tumors, RNA was extracted using the PureLink RNA Mini Kit (Invitrogen). RNA sequencing was performed by HalpoX Gene Tech (Hong Kong) on a Novaseq Xplus. Reads were aligned to the *Mus musculus* reference genome (CRCm38/mm10 assembly) using the Burrows-Wheeler aligner BWA-MEM 0.7.15 (github.com/lh3/bwa.git). The aligned reads were converted to BAM files using Picard tools 2.9.0 (broadinstitute.github.io/picard). BAM files were then processed with StringTie (ccb.jhu.edu/software/stringtie) and edgeR (bioconductor.org/packages/edgeR). Differentially expressed genes (DGEs) were defined on an absolute log2 (foldchange) ≥0.6 and FDR <0.05, unless stated otherwise.

## Whole-exome sequencing, sequence alignment, processing, and quality control

DNA extraction from formalin-fixed paraffin-embedded (FFPE) tissue and WES of mouse tumors was performed by CeGaT (Germany). Libraries were prepared using the Twist Human Core Exome kit with RefSeq and Mitochondrial panel (Twist Bioscience) and sequenced on the Illumina platform to depths of approximately 12 Gb/sample. Sequence reads were assessed for quality control using FastQC. FASTQ files were aligned to the *Mus musculus* reference genome (GRCm38/mm10 assembly) using the Burrows-Wheeler aligner BWA-MEM 0.7.15 with standard settings, and converted to BAM files using Picard tools 2.9.0. Following the alignment, we used Picard to mark duplicates. Collection of alignment and coverage metrics was performed with samtools and Picard. Targeted bases were sequenced to a mean depth of 100, and >75% of targeted bases were sequenced to 30x coverage or higher. The aligned sequence reads were visualized with the Integrative Genomics Viewer (igv.org). For the detection of single nucleotide variants (SNVs) and small insertions and deletions (indels) a Galaxy pipeline (bcftools mpileup/call; usegalaxy.eu) was used. Identified variants were annotated using Ensembl's Assembly Converter and Variant Predictor tools (ensembl.org).

## Chromatin immunoprecipitation (ChIP) assay and ChIP sequencing

ChIP assays were carried out as previously described (Simón-Carrasco et al, 2017). For precipitation of endogenous CIC proteins, 2.4 μg anti-CIC polyclonal antibodies (Invitrogen, PA5-83721) were added to 200 μg chromatin, followed by precipitation with 50 μl Protein A/G PLUS agarose (Santa Cruz Biotechnology, sc-2003). Immunoprecipitated chromatin was analyzed by qRT-PCR and quantified using the ΔΔCt method, with normalization to input DNA. For ChIP sequencing, cells were crosslinked by adding 1% formaldehyde at room temperature for 10–15 min. Crosslinking was quenched by adding glycine to a final concentration of 0.125 M. Fixed cells were collected by centrifugation at 1200 rpm for 5 min at

4 °C. The resulting pellet was washed three times with 1.5 mL of ice-cold PBS supplemented with cOmplete Protease Inhibitor (Roche). The crosslinked pellet was then lysed in 2 mL of LB1 buffer (Glycerol 1%, 50 mM HEPES-KOH pH 7.5, 140 mM NaCl, 0.5% NP-40, 0.25% Triton X-100, 1 mM EDTA, pH 8, cOmplete Mini). Samples were centrifuged at 2000×$g$ for 5 min at 4 °C, and the pellet was resuspended in 2 mL of LB2 buffer (10 mM Tris-HCl, pH 8, 200 mM NaCl, 0.5 mM EGTA, 1 mM EDTA pH 8, cOmplete Mini). The mixture was incubated on a rotating platform for 5 min at 4 °C, followed by centrifugation at 2000×$g$ for 5 min at 4 °C. Finally, the pellet containing isolated nuclei was resuspended in 200 µL of LB3 buffer (0.5% N-lauroyl sarcosine, 10 mM Tris-HCl pH 8, 100 mM NaCl, 0.1% sodium deoxycholate, 0.5 mM EGTA, 1 mM EDTA pH 8, cOmplete Mini). Samples were disrupted by sonication using a Bioruptor Plus (Diagenode) for 20 cycles of 30 s ON and 30 s OFF. After sonication, samples were centrifuged at 14,000×$g$ for 20 min at 4 °C. The supernatant was collected and incubated with 75 µL of Dynabeads Protein A (Thermo Fisher Scientific), previously conjugated with 10 µg of anti-CIC polyclonal antibody (Invitrogen, PA5-83721), for 16 h on a rotator. Beads were collected using a magnetic rack and washed six times with RIPA buffer (50 mM HEPES-KOH, pH 7.6, 0.5 M LiCl, 1% NP-40, 0.7% sodium deoxycholate, 1 mM EDTA, cOmplete Mini) at 4 °C on a rotator for 5 min per wash. Subsequently, 200 µL of elution buffer (50 mM Tris-HCl, pH 8, 1% SDS, 1 mM EDTA), 2 µL of RNase A, and 4 µL of Proteinase K were added. To reverse crosslinking, the eluted chromatin was incubated at 65 °C for 6 h. DNA was purified by phenol-chloroform extraction and quantified using a Qubit 4 Fluorometer (Invitrogen). ChIP sequencing was carried out by HalpoX Gene Tech (Hong Kong) on a DNBSEQ. Raw fastq reads were aligned to the mouse reference genome (GRCm38/mm10 assembly, http://genome.ucsc.edu) using Burrows-Wheeler aligner BWA-MEM 0.7.15 (github.com/lh3/bwa.git) with standard settings. The aligned reads were then converted to BAM files using Picard tools 2.9.0 (http://broadinstitute.github.io/picard). Duplicate reads were removed using Picard tools. BAM files were processed with MACS (github.com/taoliu/MACS/) (Zhang et al, 2008) version 3.0.0b3 for enrichment scoring and peak calling. Peaks were called using the callpeak function in MACS, and differential binding between experimental conditions was analyzed using the bdgdiff function. BED files were imported into RStudio and annotated with the R/Bioconductor package ChIPseeker (github.com/YuLab-SMU/ChIPseeker) (Yu et al, 2015). The promoter region was defined as −1 kb to +200 bp from the transcription start site (TSS).

### Genomic datasets and analyses

RNA-Seq TPM data from TCGA-LUAD samples ($n = 57$) along with corresponding mutational and copy number alteration data were downloaded using UCSC Xena (https://xenabrowser.net/datapages/). Expression of each gene was compared between tumors with KRAS/TP53 and KRAS/TP53/CIC mutations using an unpaired Student's t-test. Resulting p values were adjusted for multiple comparisons using a Bonferroni correction as an indication of significance. Normalized expression values (TPM) for each gene were also plotted using MORPHEUS software (https://software.broadinstitute.org/morpheus, Broad Institute) as a heatmap.

**The paper explained**

**Problem**

Lung adenocarcinomas driven by KRAS oncogenes are among the most severe and lethal types of cancer. Yet, the mechanisms linking KRAS mutations and progressive allelic imbalances to tumor formation remain unclear.

**Results**

Our study shows that mutant Kras copy numbers gradually increase during tumor progression. Genetic disruption of the repressor Capicua (CIC) in Kras/Trp53 mutant mice suppresses these allelic imbalances and promotes the transformation of bronchiolar Club cells, leading to an increased tumor burden and inducing resistance to MAPK pathway inhibition. Restoring CIC repressor activity or silencing its target genes Etv4 and Etv5 decreases resistance to MAPK pathway inhibition, similar to treatment with drugs (PFK15 and Tx-1123) that selectively affect the viability of Cic-deficient tumor cells.

**Impact**

We have identified the repressor CIC as a barrier to lung tumor development in Kras/Trp53 mutant mice. The absence or mutational inactivation of CIC in lung cancer patients may have adverse consequences such as increased resistance to MAPK pathway inhibitors, and reveals new vulnerabilities that could be exploited to overcome resistance.

### Statistical analyses

Data were represented as mean ± SD. P values were calculated with the unpaired Student's t-test, one-way or two-way ANOVA tests, where indicated, using GraphPad Prism (v8.4.0) software. Survival differences were calculated using the log-rank test. Differences in Kras allele frequencies determined by FISH assays were calculated using $X^2$ tests of a previously generated contingency table of allele distributions. P values <0.05 were considered statistically significant. Animals were allocated to different groups based on their genotypes, so no randomization could be performed.

### Data availability

RNA-seq data have been deposited in the NCBI Sequence Read Archive (SRA) under the accession numbers PRJNA1101109 (cell lines) or in the Gene Expression Omnibus (GEO) under the accession number GSE303269 (mouse tumors). Whole-exome sequencing data have been deposited in the SRA under the accession numbers PRJNA1160906 (cell line) and PRJNA1290372 (FFPE tissue). ChIP-seq data have been deposited in the GEO under the accession number GSE303110. URLs: PRJNA1101109, GSE303269, PRJNA1160906, PRJNA1290372, GSE303110.

The source data of this paper are collected in the following database record: biostudies:S-SCDT-10_1038-S44321-025-00326-z.

### Peer review information

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

## Acknowledgements

We thank R. Villar and M. San Roman for technical assistance, members of the animal facilities of the CNIO and the CIC for animal care, the Histopathology Core Units of the CNIO and the CIC for tissue processing and immunostainings, and the Genomics Unit of the CNIO for mouse genotyping. This work was supported by a grant from the *Agencia Estatal de Investigación, Ministerio de Ciencia e Innovación* (grant PID2020-116705RB-I00) to M.D. M.D. also received funding from the Scientific Foundation of the Spanish Association Against Cancer (grant LABAE211678DROS). This work was further supported by a grant from the Scientific Foundation of the Spanish Association Against Cancer (*Programa Excelencia* 2022 grant EPAEC222641CICS); grants from *Instituto de Salud Carlos III*, co-funded by FEDER (grants PI23/01932 and PI21/01641) and the Scientific Foundation of the Spanish Association Against Cancer (grant LABAE20049RODR) to S.R.-P.; as well as a grant from the *Agencia Estatal de Investigación, Ministerio de Ciencia e Innovación* (grant PID2020-119248GB-I00) to G.J. T.P. was supported in part by a grant from the *Comunidad de Madrid* (grant P2022/BMD-7321/MITIC-CM). This work was also supported in part by grants from the European Research Council (ERC-AG/695566 [THERACAN]), the *Comunidad de Madrid* (B2017/BMD-3884 iLUNG-CM) and the *Agencia Estatal de Investigación, Ministerio de Ciencia e Innovación* (RTI2018-094664-B-I00) to M.B. M.B. is a recipient of an Endowed Chair from the AXA Research Fund. O.B. was a recipient of a fellowship from the *Formación de Personal Investigador* (FPI) program of the *Agencia Estatal de Investigación, Ministerio de Ciencia e Innovación* (PRE2019-091685). M.Salmón was supported by a predoctoral contract "Severo Ochoa" from the *Formación de Personal Investigador* (FPI) program of the *Agencia Estatal de Investigación, Ministerio de Ciencia e Innovación* (BES-2016-079096). A.A.F.-G. was a recipient of a CONAHCYT fellowship from the Mexican Government. L.S.-C. was supported by a Postdoctoral Research Contract from the Scientific Foundation of the Spanish Association Against Cancer (POSTD211274SIMÓ).

## Author contributions

**Irene Ballesteros-González**: Conceptualization; Formal analysis; Investigation; Visualization; Methodology; Writing—review and editing. **Iván Hernández-Navas**: Investigation; Formal analysis; Validation. **Oksana Brehey**: Investigation. **Carmen G Lechuga**: Investigation. **Marina Salmón**: Investigation. **Morena Scotece**: Investigation. **Ricardo Velasco-Vicente**: Investigation. **Alejandra A Florez-Gómez**: Investigation. **Antonio Cebriá**: Investigation. **Lucía Simón-Carrasco**: Conceptualization; Investigation; Writing—review and editing. **Gerardo Jiménez**: Conceptualization; Funding acquisition; Writing—review and editing. **Monica Musteanu**: Resources; Methodology. **Carmen Guerra**: Resources; Methodology. **Orlando Domínguez**: Formal analysis; Investigation; Methodology. **Eduardo Caleiras**: Formal analysis; Investigation. **Carmen Blanco-Aparicio**: Formal analysis; Investigation. **Tirso Pons**: Data curation; Formal analysis; Investigation; Methodology. **Irene Ferrer**: Formal analysis; Investigation. **Luis Paz-Ares**: Resources; Methodology. **Raul Torres-Ruiz**: Investigation. **Sandra Rodríguez-Perales**: Formal analysis; Funding acquisition; Investigation; Visualization. **Mariano Barbacid**: Conceptualization; Resources; Funding acquisition. **Matthias Drosten**: Conceptualization; Formal analysis; Supervision; Funding acquisition; Investigation; Visualization; Methodology; Writing—original draft; Writing—review and editing.

Source data underlying figure panels in this paper may have individual authorship assigned. Where available, figure panel/source data authorship is listed in the following database record: biostudies:S-SCDT-10_1038-S44321-025-00326-z.

## Disclosure and competing interests statement

The authors declare no competing interests.

# Expanded View Figures

**Figure EV1.   Tumor burden in KP mice expressing WT CIC or CIC$^{\Delta 2-6}$.**

(**A**) Quantification of lesions in KP (blue bar, $n = 8$) or KPCic mice (red bar, $n = 8$) infected with $5 \times 10^5$ pfu Ad-Cre at a humane endpoint. Results are shown as mean ± SD. Statistics, unpaired *t*-test. (**B**) Quantification of tumor grades (G1–G5) in lesions from KP (blue bars, $n = 8$) or KPCic mice (red bars, $n = 8$) infected with $5 \times 10^5$ pfu Ad-Cre at a humane endpoint. Results are shown as mean ± SD. Statistics, multiple *t*-tests. (**C**) Quantification of HMGA2+ tumors in KP (blue bar, $n = 3$) or KPCic mice (red bar, $n = 4$) infected with $5 \times 10^5$ pfu Ad-Cre at a humane endpoint. Results are shown as mean ± SD. Statistics, unpaired *t*-test. (**D**) Representative images of HMGA2 immunostaining in lung sections from a KP and a KPCic mouse infected with $5 \times 10^5$ pfu Ad-Cre at a humane endpoint. Scale bar, 2 mm. (**E**) SPC, TTF-1, and HMGA2 immunostaining in consecutive sections from a lung tumor obtained from a KP mouse. Scale bar, 200 µm. (**F**) Quantification of tumor grades (G1–G5) in lesions from KP (blue bars, $n = 10$) or KPCic mice (red bars, $n = 11$) infected with $5 \times 10^7$ pfu Ad-Cre 5 months after infection. Results are shown as mean ± SD. Statistics, multiple *t*-tests. (**G**) Quantification of HMGA2+ tumors in KP (blue bar, $n = 3$) or KPCic mice (red bar, $n = 3$) infected with $5 \times 10^7$ pfu Ad-Cre 5 months after infection. Results are shown as mean ± SD. (**H**) Representative images of HMGA2 immunostaining in lung sections from a KP and a KPCic mouse infected with $5 \times 10^7$ pfu Ad-Cre 5 months after infection. Arrowheads show positive staining. Scale bar, 2 mm.

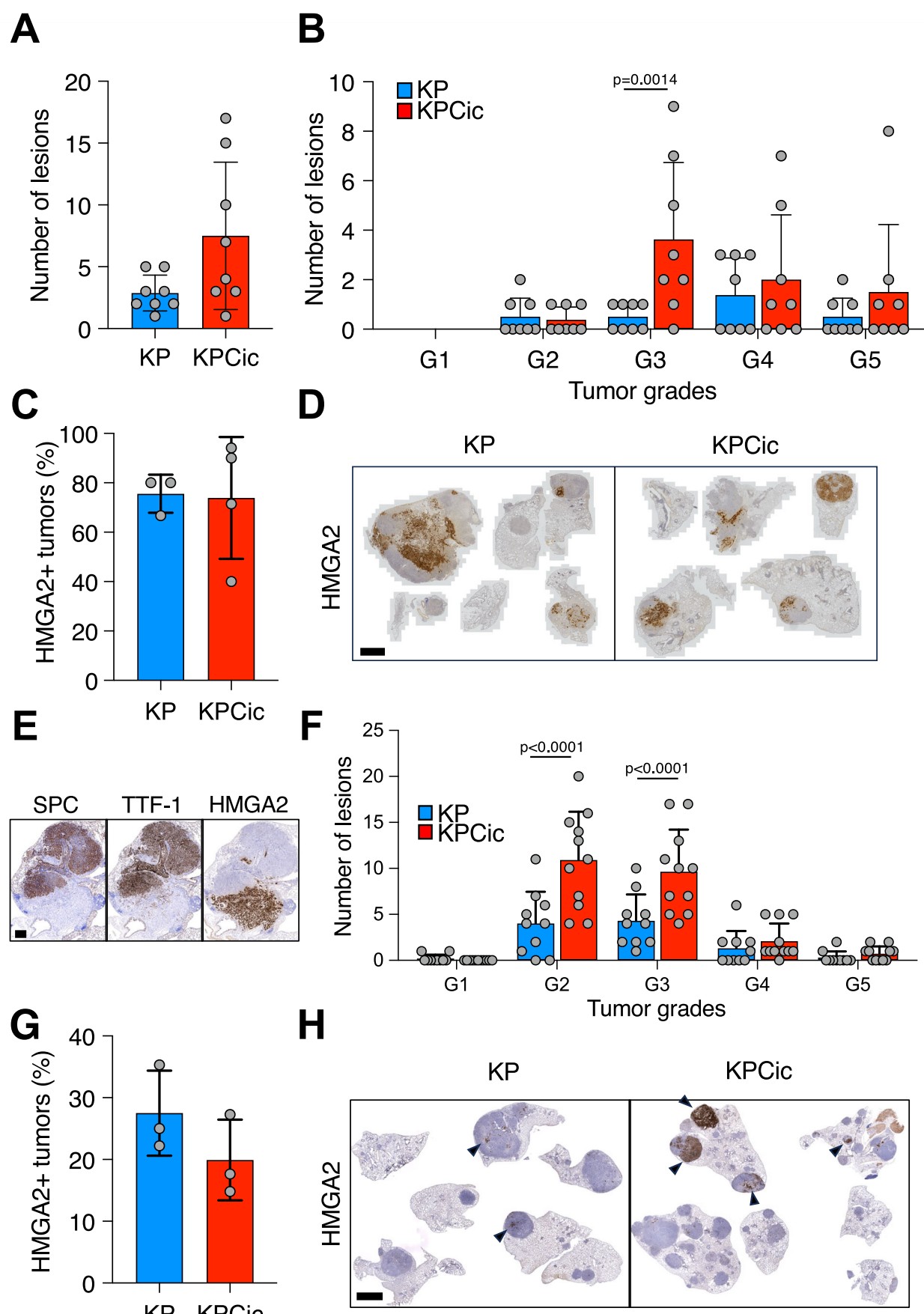

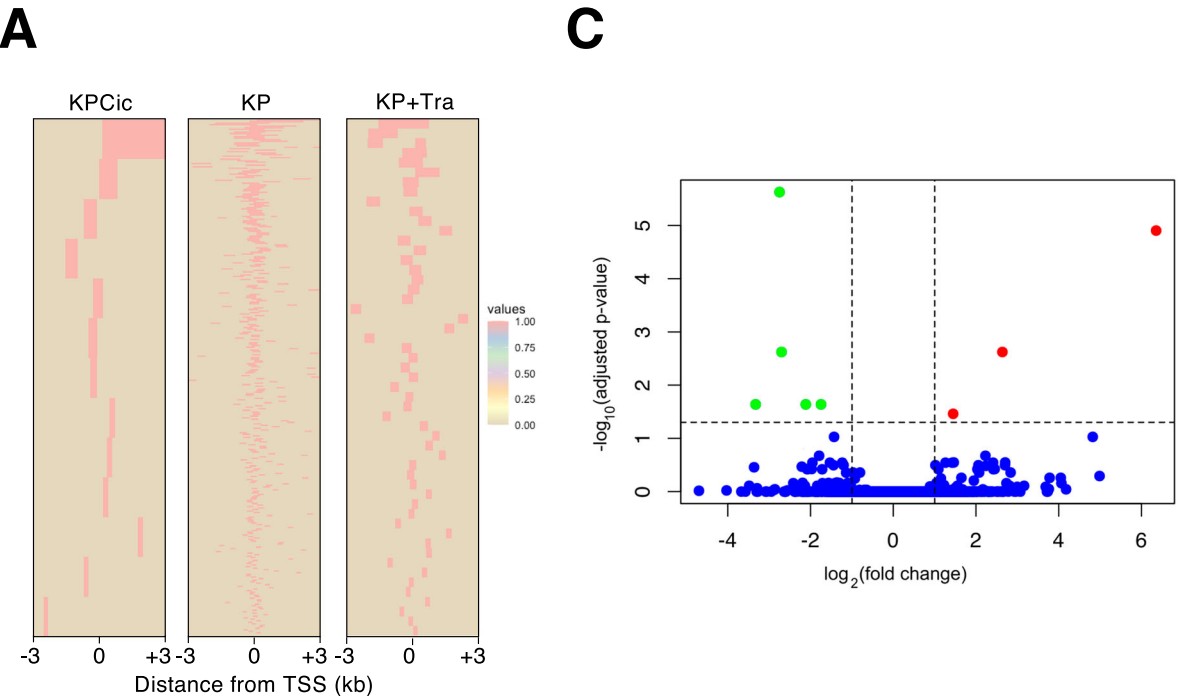

**Figure EV2.   ChIP- and RNA-seq analyses of KP and KPCic tumors.**

(A) Heatmap showing the distribution of CIC peak distances to the nearest transcriptional start site as determined by ChIPseeker in KPCic cells, KP cells and KP cells treated with 100 nM trametinib for 24 h. (B) ChIP-seq normalized coverage of CIC binding to representative promoters in KPCic cells, KP cells and KP cells treated with 100 nM trametinib for 24 h. The Y-axis values indicate the mean of normalized reads per 10 bp, using BPM with bamCoverage of deepTools. (C) Volcano plot of gene expression changes in KP ($n = 5$) vs. KPCic ($n = 5$) lung tumors obtained 5 months after infection with Ad-Cre. Statistical analysis was performed using the likelihood ratio test as implemented in the Bioconductor R package edgeR. To control the false discovery rate, $p$ values were adjusted using the Benjamini–Hochberg method.

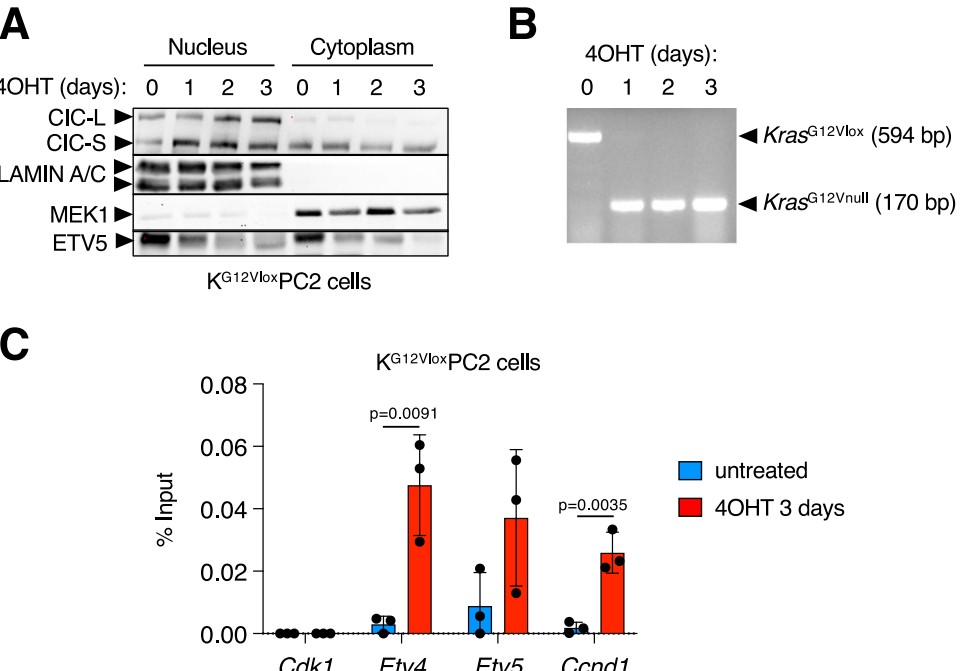

**Figure EV3.   Control of CIC activity by oncogenic KRAS signaling in lung cancer cells**

(A) Western blot analysis of CIC and ETV5 expression in cytoplasmic and nuclear fractions of K$^{G12Vlox}$PC2 cells treated with 4-hydroxytamoxifen (4OHT) for the indicated time. MEK1 expression served as a marker for the cytoplasmic fraction and Lamin A/C expression for the nuclear fraction. (B) PCR analysis to confirm excision of the Kras$^{G12Vlox}$ allele in K$^{G12Vlox}$PC2 cells treated with 4OHT for the indicated time. Kras$^{G12Vlox}$ (594 bp) and Kras$^{G12Vnull}$ (170 bp) alleles are indicated. (C) Chromatin immunoprecipitation assay using CIC antibodies in untreated K$^{G12Vlox}$PC2 cells (blue bars) and K$^{G12Vlox}$PC2 treated with 4OHT for 3 days (red bars). Binding to CIC-binding sites (CBS) in the Etv4, Etv5 and Ccnd1 promoters as well as the Cdk1 promoter which lacks CBS was analyzed by qRT-PCR and normalized to the amount of input DNA. Results are shown as mean ± SD. Statistics, unpaired t-test ($n = 3$ biological replicates).

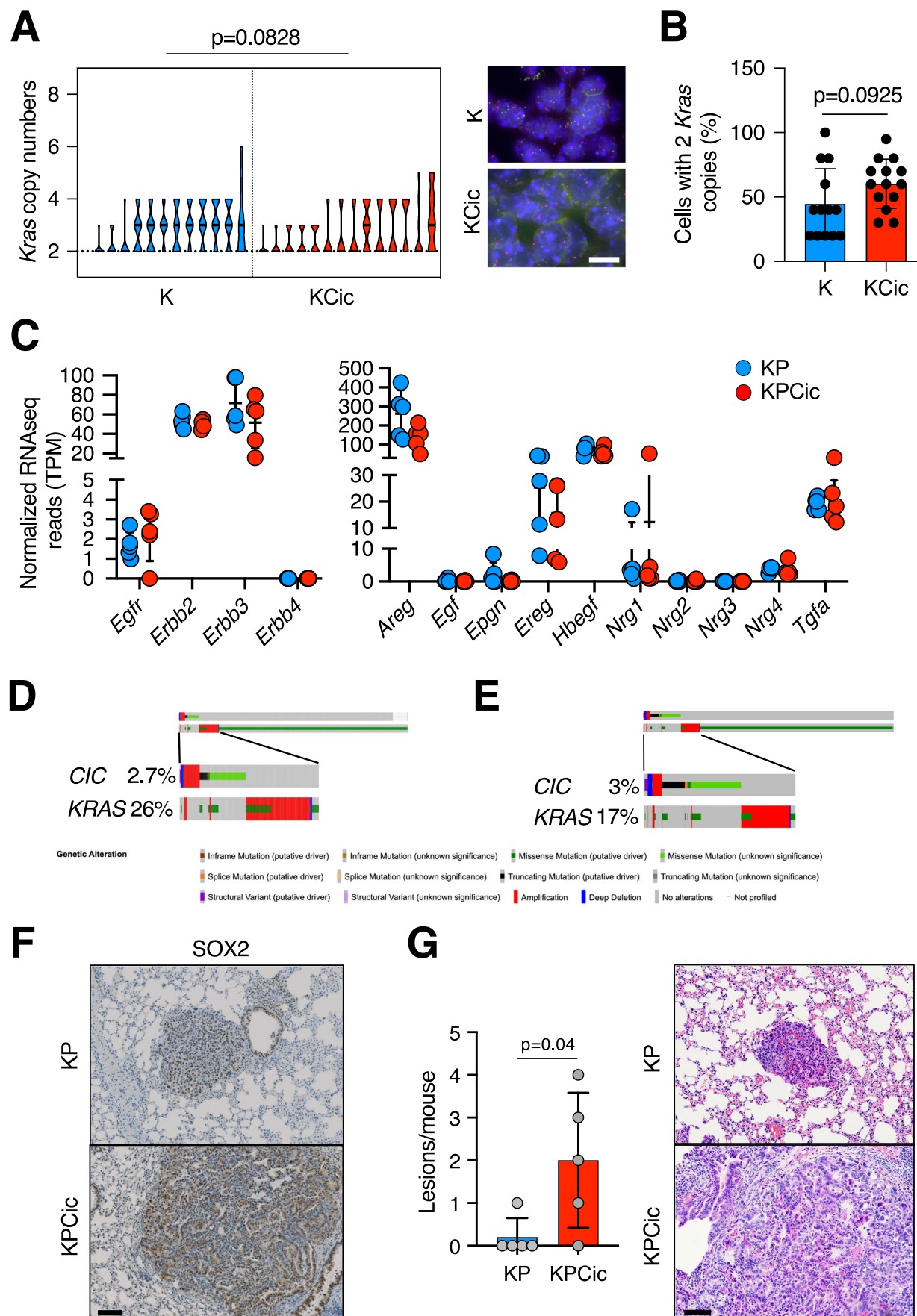

**Figure EV4. Characterization of the impact of *Cic* deletion.**

(A) Left, quantification of *Kras* copy numbers by FISH staining in sections from lung tumors in K ($n = 12$ tumors, $n = 3$ mice; blue symbols) and KCic ($n = 14$ tumors, $n = 3$ mice, red symbols) mice obtained 10–12 months after infection with Ad-Cre. Statistics, $X^2$ test. Right, representative interphase FISH images of tumors obtained from K and KCic mice 10–12 months after infection with Ad-Cre. Red, *Kras* probe. Green, chromosome 6 probe. Scale bar, 5 μm. (B) Quantification of tumor cells retaining *Kras* 2 N in samples from (A). Statistics, unpaired *t*-test. (C) Normalized RNA-seq reads of the indicated genes from five independent KP as well as KPCic tumors obtained 5 months after infection with Ad-Cre. Results are shown as mean ± SD. Statistics, multiple *t*- tests. (D) Mutations in *CIC* and *KRAS* in human LUAD samples obtained from the TCGA database. (E) Mutations in *CIC* and *KRAS* in human Pan-Cancer samples obtained from the TCGA database. (F) Representative SOX2 IHC stainings in tumors from KP or KPCic mice 5 months after infection with Ad-CC10-Cre. Scale bar, 100 μM. (G) Left: Quantification of tumors in KP ($n = 5$) and KPCic ($n = 5$) mice 4 months after infection with Ad-CC10-Cre. Results are shown as mean ± SD. Statistics, unpaired *t*-test. Right, representative H&E images of tumors from KP or KPCic mice 4 months after infection with Ad-CC10-Cre. Scale bar, 100 μM.

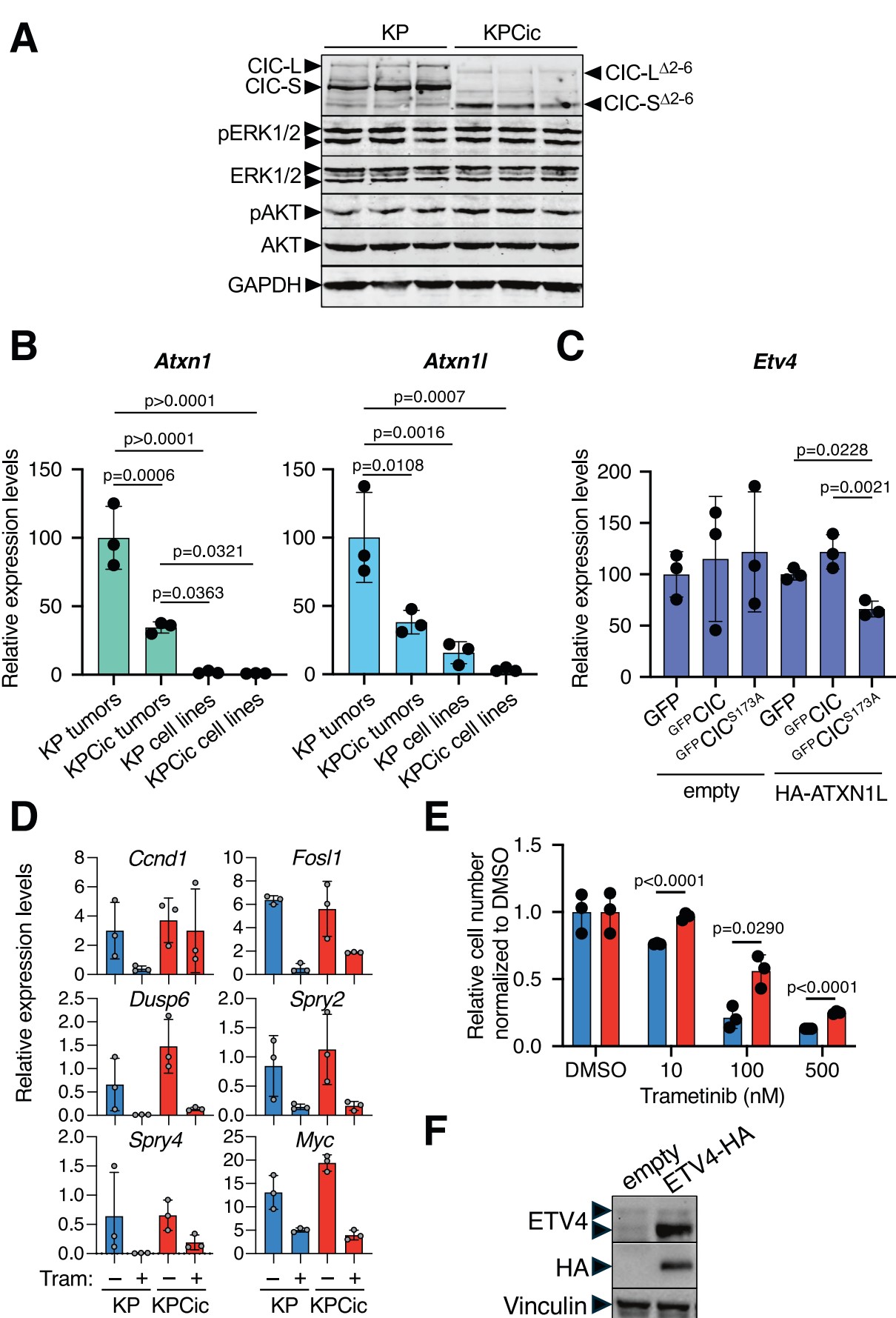

**Figure EV5.   Analysis of signaling pathways in KP and KPCic cell lines.**

(A) Western blot analysis of CIC, pERK1/2, ERK1/2, pAKT and AKT expression in KP and KPCic cell lines. GAPDH expression served as a loading control. (B) qRT-PCR of *Atxn1* and *Atxn1l* expression in KP tumors ($n = 3$), KPCic tumors ($n = 3$), KP cell lines ($n = 3$), and KPCic cell lines ($n = 3$). Results are shown as mean ± SD. Statistics, one-way ANOVA ($n = 3$ technical replicates). (C) qRT-PCR of *Etv4* expression in a KPCic cell line either infected with empty lentiviruses or lentiviruses stably expressing HA-ATXN1L 72 h after infection with Ad-GFP, Ad$^{GFP}$CIC or Ad$^{GFP}$CIC$^{S173A}$. Results are shown as mean ± SD. Statistics, one-way ANOVA ($n = 3$ technical replicates). (D) qRT-PCR of the indicated genes in KP (blue bars, $n = 3$) and KPCic cell lines (red bars, $n = 3$) treated with DMSO (–) or 20 nM trametinib (Tram) for 24 h (+). Results are shown as mean ± SD. (E) Proliferation of a KP cell line infected with empty retroviruses (blue bars) or pLPC-ETV4-HA (red bars), 4 days after treatment with DMSO, 10 nM trametinib, 100 nM trametinib or 500 nM trametinib. Results are shown as mean ± SD. Statistics, multiple *t*-tests ($n = 3$ technical replicates). (F) Western blot analysis of ETV4-HA expressing using ETV4 and HA antibodies in KP cells a KP cell line infected with empty retroviruses or pLPC-ETV4-HA. Vinculin expression served as a loading control.

                                          