## [Peer Review File · EMBO Molecular Medicine]

The repressor Capicua is a barrier to lung tumor development driven by Kras/Trp53 mutations

Irene Ballesteros-González, Iván Hernández-Navas, Oksana Brehey, Carmen Lechuga, Marina Salmón, Morena Scotece, Ricardo Velasco-Vicente, Alejandra Florez-Gómez, Antonio Cebriá, Lucía Simón-Carrasco, Gerardo Jiménez, Monica Musteanu, Carmen Guerra, Orlando Domínguez, Eduardo Galeiras, Carmen Blanco-Aparicio, Tirso Pons, Irene Ferrer, Luis Paz-Ares, Raul Torres, Sandra Rodríguez-Perales, Mariano Barbacid, and Matthias Drosten

Corresponding author: Matthias Drosten (mdrosten@usal.es)

Review Timeline:

Submission Date:	17th Feb 25
Editorial Decision:	19th Mar 25
Revision Received:	10th Aug 25
Editorial Decision:	1st Sep 25
Revision Received:	2nd Oct 25
Accepted:	9th Oct 25

Editor: Lise Roth

Transaction Report:

19th Mar 2025

Dear Dr. Drosten,

Thank you for the submission of your manuscript to EMBO Molecular Medicine. We have now received feedback from the three reviewers who agreed to evaluate your manuscript. As you will see from the reports below, the referees acknowledge the interest of the study and are overall supporting publication of your work pending appropriate revisions.

Addressing the reviewers' concerns in full will be necessary for further considering the manuscript in our journal, and acceptance of the manuscript will entail a second round of review. EMBO Molecular Medicine encourages a single round of revision only and therefore, acceptance or rejection of the manuscript will depend on the completeness of your responses included in the next, final version of the manuscript. For this reason, and to save you from any frustrations in the end, I would strongly advise against returning an incomplete revision.

We are expecting your revised manuscript within three to four months, if you anticipate any delay, please contact us.

We require:

4) A .docx formatted letter INCLUDING the reviewers' reports and your detailed point-by-point responses to their comments. As part of the EMBO Press transparent editorial process, the point-by-point response is part of the Review Process File (RPF), which will be published alongside your paper.

5) A complete author checklist, which you can download from our author guidelines (<https://www.embopress.org/page/journal/17574684/authorguide#submissionofrevisions>). Please insert information in the checklist that is also reflected in the manuscript. The completed author checklist will also be part of the RPF.

6) All Materials and Methods need to be described in the main text using our 'Structured Methods' format. According to this format, the Methods section includes a Reagents and Tools Table (listing key reagents, experimental models, software and relevant equipment and including their sources and relevant identifiers) followed by a Methods and Protocols section describing the methods, ideally using a step-by-step protocol format. The aim is to facilitate adoption of the methodologies across labs. Please download and fill our Reagents and Tools Table template (.docx), which you can find in our author guidelines:

<https://www.embopress.org/doi/10.15252/msb.20178071>

7) Please note that all corresponding authors are required to supply an ORCID ID for their name upon submission of a revised manuscript.

8) It is mandatory to include a 'Data Availability' section after the Materials and Methods. Before submitting your revision, primary datasets produced in this study need to be deposited in an appropriate public database, and the accession numbers and database listed under 'Data Availability'. Please remember to provide a reviewer password if the datasets are not yet public (see <https://www.embopress.org/page/journal/17574684/authorguide#dataavailability>).

9) For data quantification: please specify the name of the statistical test used to generate error bars and P values, the number (n) of independent experiments (specify technical or biological replicates) underlying each data point and the test used to calculate p-values in each figure legend. The figure legends should contain a basic description of n, P and the test applied. Graphs must include a description of the bars and the error bars (s.d., s.e.m.). Please provide exact p values.

10) Our journal encourages inclusion of *data citations in the reference list* to directly cite datasets that were re-used and obtained from public databases. Data citations in the article text are distinct from normal bibliographical citations and should directly link to the database records from which the data can be accessed. In the main text, data citations are formatted as follows: "Data ref: Smith et al, 2001" or "Data ref: NCBI Sequence Read Archive PRJNA342805, 2017". In the Reference list, data citations must be labeled with "[DATASET]". A data reference must provide the database name, accession number/identifiers and a resolvable link to the landing page from which the data can be accessed at the end of the reference. Further instructions are available at .

11) We replaced Supplementary Information with Expanded View (EV) Figures and Tables that are collapsible/expandable online. EV Figures should be cited as 'Figure EV1, Figure EV2' etc... in the text and their respective legends should be included in the main text after the legends of regular figures.

12) The paper explained: EMBO Molecular Medicine articles are accompanied by a summary of the articles to emphasize the major findings in the paper and their medical implications for the non-specialist reader. Please provide a draft summary of your article highlighting

13) Author contributions: CRedit has replaced the traditional author contributions section because it offers a systematic machine readable author contributions format that allows for more effective research assessment. Please remove the Authors Contributions from the manuscript and use the free text boxes beneath each contributing author's name in our system to add specific details on the author's contribution. More information is available in our guide to authors.

Please also suggest a visual abstract to illustrate your article as a PNG file 550 px wide x 300-600 px high. A cropped portion of this image will serve as thumbnail for the table of content on our webpage.

16) As part of the EMBO Publications transparent editorial process initiative (see our Editorial at <http://embomolmed.embopress.org/content/2/9/329>), EMBO Molecular Medicine will publish online a Review Process File (RPF) to accompany accepted manuscripts.

In the event of acceptance, this file will be published in conjunction with your paper and will include the anonymous referee reports, your point-by-point response and all pertinent correspondence relating to the manuscript. Let us know whether you agree with the publication of the RPF and as here, if you want to remove or not any figures from it prior to publication. Please note that the Authors checklist will be published at the end of the RPF.

I look forward to receiving your revised manuscript.

Yours sincerely,

Lise Roth

***** Reviewer's comments *****

Referee #1 (Comments on Novelty/Model System for Author):

The model systems used are appropriate

Referee #1 (Remarks for Author):

The manuscript by Ballesteros-Gonzales et al uses sporadic CRE-dependent allelic mice to examine the role of Capicua (Cic) loss in lung cancer driven by activating mutation of KRas concomitant with loss of p53. The authors show convincingly that loss of Cic in the KP background increases the number of lung tumours with a resulting decrease in overall lifespan. Interestingly, loss of Cic appears to obviate the requirement for increased Ras->Erk pathway activity as tumours progress and reduces the overall level of allelic imbalance observed in the model. The authors go on to show that upregulation of Ets transcription factor expression appears to largely mediate the impact of Cic mutation, while CIC mutation in human LuAd correlates strongly with high expression of ETS TFs. They finally go on to show that loss of Cic can combine with loss of P53 to drive tumours in a subset of mice lacking KRas mutation. This is an interesting study and the observations are certainly novel. The manuscript is well-written and the figures are well laid out and easy to follow. Some of the conclusions however are rather overstated and some relatively minor revisions would likely improve the final paper.

Major Points:

- 1) The authors provide a detailed analysis of KRasG12V amplification as a means to explain the progressive increase in p-ERK signal measured in KP tumours but reduced in KPCic tumours. Kruspig et al (2018) provided an alternative mechanism for increased flux through the Ras->Erk pathway through spontaneous upregulation of ERBB/EGFR ligands, especially AREG and EREG. Indeed, this route appears to be much more commonly used than KRas copy number gain in human KRas mutant LuAd. Is this route to tumour progression also suppressed upon loss of Cic?
- 2) Is loss of p53 required to expose the tumour suppressive activity of Cic? The authors have presumably examined tumour formation in KCic mice that would have arisen during breeding to generate the more complex KPCic combination - some comment (or ideally data) on this should be included.
- 3) The statement in the introduction that loss of Cic is a "prerequisite for efficient tumour formation" is poorly supported. Only 1 genetic model is examined and, arguably more importantly, Cic is mutated in only a small subset of human LuAd, with the majority of these being mutations of unknown impact.

Additional points:

- 1) In Figure 1, RNA-SEQ comparison of tumours from KP vs KPCic mice would likely be more informative of the transcriptional impact of Cic loss in situ. I appreciate this was done for cell lines, however, cell lines can diverge considerably from their parent tumours and moreover fail to incorporate the impact on the tumour microenvironment.
- 2) From the data in figure 1 the authors conclude that loss of Cic facilitates tumour initiation "but does not contribute to tumour progression". I don't believe this statement is fully justified given that a) the KP model is already aggressive from initiation; b)

- they have not shown any data on KCic mice and c) no other allelic combination has been examined (aside from the partially penetrant PCic combination). Cic may well play a role in early tumour progression that is bypassed by genetic deletion of p53
- 3) From the human mutation data shown in Fig S2, alterations in CIC appear to be mutually exclusive with KRAS mutation. The very small number of KRAS mutant tumours that do show CIC alteration have mutations of unknown consequence in the latter.
 - 4) The data shown in Figure 3 using Ad-CC10-CRE to target allelic activation in Club cells do not offer a "likely explanation for the increased tumour burden in KPCic lungs", contrary to what is stated. On the contrary, no actual tumours are shown to arise - only areas of dysplastic tissue that likely represent pseudo-oncogenic dead-ends.
 - 5) In the Cic restoration to cell line experiments, the level of GFP-Cic expression relative to endogenous Cic is not discernible from the blots shown in 4C.
 - 6) Please include the Z-score range for Figure 5A
 - 7) In Figure 5, Ets proteins remain strikingly sensitive to Trametinib in the KPCic cell lines despite the apparent resilience of Ets transcripts. While depletion of Ets from KPCic lines does restore a degree of sensitivity to Trametinib, in one of the lines depletion of Ets4/5 alone decimates colony formation in the absence of MEK inhibition. Does overexpression of Ets render KP or H358 cells resistant to Trametinib? The data at present do not fully support the headline that Ets mediate resistance. The loss of proliferation in human cells depleted of ETS may be entirely unrelated to CIC status.
 - 8) The final statement of the results section ("...CIC is a crucial effector of KRAS pathway in LuAD") does not make sense.

Referee #2 (Remarks for Author):

This manuscript by Ballestros-Gonzales et al. document the important role of CIC in suppressor KRAS-driven lung cancer. They assess GEMMs, perform cell lines and molecular studies, as well as assess drug responses. I found the discuss enjoyable to read with clear thought about how their works fits into the existing literatures as well as an articulation of their proposed model.

Major/Moderate Comments:

1. Figure 2B. I do not find the plot very intuitive. Could there just be a dot for each tumor. I presume that data is per tumor, not per mouse. So N=6 mice etc is not as helpful as presenting N=how many tumors. Actually, I see now that each column is a mouse. In that case the data seems quite weird, why would all tumors in some mice be 2 copies while in other mice the majority of tumors have >3 copies. I guess it's nice to plot per mouse somewhere, but perhaps they can just bundle all tumors across all mice and then make comparisons. This seems like a very important comparison.
2. While the authors propose the CIC inactivation enable enhance tumorigenesis from CC10+ cells. I do not see the data to support this. Are there more tumors and/or larger tumors in Aden-CC10-Cre KPCic mice relative to KP mice? There is one tumor shown in Fig 3C, but is there really more big tumors? They should show tumor burden like in Figure . Are they confident that this is THE reason, or could this be ONE reason for the effect of CIC inactivation? Also, they do not provide any evidence the CIC isn't a tumor suppressor when tumor arrive from AT2 cells. Again, I am not saying that they should do that be rather be careful about their claims.
3. For cell line studies in Figures 4-6 the authors should make sure that all data is from multiple cell lines (n=3+ would be best). Maybe they don't need to repeat every experiment but a few more of the critical ones should be repeated in multiple KP and KPCic lines.
4. The development of tumors in CIC/p53 mice is a very interesting result as LUADs without driving oncogenes represent a large fraction of lung tumors. I think that this analysis could be improved by providing addition high resolution images, getting assess from a pathologist, maybe HMG2 staining. The variability across mice is concerning so maybe just some more data to convince the reader that these are really lung tumors (were they able to generate a cell line? I would hope they tried as they are p53-deficient tumors). For the analysis of KRAS mutations they should also look for Q61 mutations, and maybe for BRAF and EGFR mutations. I would not suggest looking harder than that as they likely have other genomic changes but the major point here would be that they lack canonical LUAD oncogenes.
5. While the impact of CIC inactivation significant, this is nowhere as big of an effect as inactivating many other tumor suppressor (e.g Lkb1). In particular the effect of inactivating NF1 (Wang et al. EmboMM 2019) is much larger. Thus, it seems unlikely that THE effect of increased MAPK is to inactivate CIC. Not sure if the authors want to mention this.

Minor Comments:

6. Stating the "CIC inactivation is a prerequisite for efficient tumor development" is a little too strong. It's just not quite the right word
7. The conclusion about increased Grade 3 tumors at end point is pretty weak, not sure if they really want to claim that. Most people would present that fraction of tumor of each grade, otherwise it more about tumor number and less about progression. Regardless of what they decide to claim regarding Grade there doesn't really seem to be a difference in progression

8. The IHC in Figure 1E doesn't seem representative of the data that is quantified to the right.
9. The absence of any data comparing K to KCic mice is striking. The authors might want to mention something about this in the Discussion. Whether the impact of CIC inactivation also exists in the KRAS/p53-proficient tumors remains unknown.
10. If they want to claim that pERK correlates with Kras copy number, then they need to plot on a per tumor level, not just a correlation across time points.
11. Figure 2G is great. But are 2F and 2G just the same data plotted two different ways? If so, then remove 2F.
12. It really is easier to review manuscripts that have line numbers and the legends on the figures, but I managed.
13. The running title doesn't seem quite right

Referee #3 (Remarks for Author):

The authors determined the role of Capicua (CIC) in lung tumorigenesis using GEMM. This work deserves publication. I have some comments, which may make this paper better:

- 1) Have the authors made KCic (no p53 deletion) mice and compared to K or KP mice for Fig. 1? Based on Fig. 7 data, would KCic be similar to K? Is CIC dispensable as a tumor suppressor for Kras mutant lung cancer in the presence of intact p53? It would be good to show the data and/or include in the text.
- 2) Etv4 and Etv5 as downstream targets for CIC have been reported by others (Okimoto 2017 and Park 2017). It would be better to report new genes regulated by CIC that are critical for lung tumor growth. RNA-seq using the samples from Fig 1G may help. The authors should also do ChIP-seq instead of ChIP to demonstrate a better mechanism by which CIC regulates downstream genes. Ccnd1 mRNA data in Fig. 1G doesn't seem to be consistent with ChIP data at Ccnd1 locus in Fig. S2C. RNA-seq and ChIP-seq will provide a better and unbiased understanding of CIC as a transcriptional repressor and the lung cancer field would appreciate the effort.
- 3) For Fig. 3C, I recommend looking at SOX2 expression to ensure the author's claim that Cic deletion accelerates the transformation of Club cells into ATII-like tumor cells. I wonder if SOX2 expression is maintained or lost in the KPCic tumors. SOX2 staining information is available at Sutherland et al (2014).
- 4) In Fig. 4, does CIC or CICS173A function without ATXN1L in the in vitro cell line experiments? The authors should show the data as to the function of CIC and CICS173A without ATXN1L in the in vitro cell line. Did these cell lines lose ATXN1L after they were isolated from lung tumors? If so, it's good to show the data (RNA and/or protein) as well (in vitro vs in vivo) and mention these data in the text.

Response to reviewers:

We thank the reviewers for their constructive suggestions to improve our manuscript. Below, we provide a point-by-point response to their comments.

Referee #1:

1. The authors provide a detailed analysis of *KRas*G12V amplification as a means to explain the progressive increase in p-ERK signal measured in KP tumours but reduced in KPCic tumours. Kruspig et al (2018) provided an alternative mechanism for increased flux through the Ras->Erk pathway through spontaneous upregulation of ERBB/EGFR ligands, especially AREG and EREG. Indeed, this route appears to be much more commonly used than *KRas* copy number gain in human *KRas* mutant LuAd. Is this route to tumour progression also suppressed upon loss of *Cic*?

We thank the reviewer for this comment. To address this question, we have analyzed the expression levels of ERBB/EGFR ligands as mentioned in Kruspig et al (2018) in a new RNA-seq dataset obtained from KP and KPCic lung tumors (see point 4 below). In general, we observed no significant differences in the expression levels of any of these genes between KP and KPCic tumors, suggesting that this route to tumor progression is not suppressed upon loss of *Cic*. We have included this information to the new Figure EV4C and the text on pages 10/11 (results section).

Fig. EV4C

2. Is loss of *p53* required to expose the tumour suppressive activity of *Cic*? The authors have presumably examined tumour formation in KPCic mice that would have arisen during breeding to generate the more complex KPCic combination - some comment (or ideally data) on this should be included.

This is an excellent point raised by the reviewer (see also point 9, reviewer 2 and point 1, reviewer 3). Unfortunately, it was always our intention to perform this study in the context of *p53* loss since *Kras*^{G12V} activation in a *p53* WT context results in relatively slow tumor progression rates and would have increased the number of required mice considerably. Nevertheless, during the early phases of generating KP and KPCic strains, we obtained a few K as well as KPCic animals with WT *p53*. We infected a low number of these animals (3+3) with Ad-Cre and collected samples when these animals became sick (between 10 and 12 months). We have now used these samples to analyze *Kras* copy numbers by FISH and, although the difference does not quite reach statistical significance ($p=0.0828$, χ^2), there is also a tendency towards higher *Kras* copy numbers in K mice. When only comparing the percentage of tumor cells retaining 2 *Kras* copies, we also observed a trend towards higher percentages in KPCic mice ($p=0.09$). However, since statistical significance is not reached, we concluded that absence of *Cic* does not significantly suppress *Kras* copy number expansion when *p53* is retained. These results were included in the manuscript as the new Figs. EV4A+B including a statement in the results section on page 10 (“...this suppression was not as evident in tumors retaining functional *p53*.”) and a statement in the discussion section on page 18. There, we reasoned that lack of statistically significant differences could be a consequence of the relative old age of these animals (10-12 months after infection). Moreover, since our results in fact only expose *Cic*'s

tumor suppressor activity in the KP model, we propose to change the title of our manuscript to: “*The repressor Capicua is a barrier to lung tumor development driven by KRAS/TRP53 mutations.*”

Figs. EV4A+B

3. The statement in the introduction that loss of *Cic* is a "prerequisite for efficient tumour formation" is poorly supported. Only 1 genetic model is examined and, arguably more importantly, *Cic* is mutated in only a small subset of human LuAd, with the majority of these being mutations of unknown impact.

We agree that our results do not really support this general claim. Therefore, we have now changed the phrase to: "...and reveal that *CIC* inactivation enhances tumor development." (page 7)

4. In Figure 1, RNA-SEQ comparison of tumours from KP vs KPCic mice would likely be more informative of the transcriptional impact of *Cic* loss in situ. I appreciate this was done for cell lines, however, cell lines can diverge considerably from their parent tumours and moreover fail to incorporate the impact on the tumour microenvironment.

We agree with the reviewer that profiling the transcriptome of lung tumors would be more informative since the transcriptome of cell lines derived from these tumors has likely suffered changes. As suggested, we have now performed RNA-seq of 5 new KP as well as KPCic tumors that were extracted 5 months after infection. Unexpectedly, only very few genes were significantly altered (see Dataset EV1). While in our previous Fig. 1G we detected significant differences in *ETV4* and *ET5* expression, these differences could not be recapitulated by RNA-seq. Based on this RNAseq analysis, we concluded that: "...once tumors have formed in the absence of *CIC*, they are largely indistinguishable from those in KP mice." (page 9). We have removed the original Fig. 1G since we could not recapitulate the significant increase in *Etv4* and *Etv5* expression and added a volcano plot summarizing the data as the new Fig. EV2C. We have also added a brief section to the discussion (page 18).

Fig. EV2C

5. From the data in figure 1 the authors conclude that loss of *Cic* facilitates tumour initiation "but does not contribute to tumour progression". I don't believe this statement is fully justified given that a) the KP model is already aggressive from initiation; b) they have not shown any data on KCic mice and c) no other allelic combination has been examined (aside from the partially

penetrant PCic combination). Cic may well play a role in early tumour progression that is bypassed by genetic deletion of p53

We thank the reviewer for pointing this out. Since we only studied the contribution of CIC to tumor formation in KP mice, we have now modified this phrase on page 9 to: "...but does not notably contribute to tumor progression in KP mice."

6. From the human mutation data shown in Fig S2, alterations in CIC appear to be mutually exclusive with KRAS mutation. The very small number of KRAS mutant tumours that do show CIC alteration have mutations of unknown consequence in the latter.

As indicated in Fig. 7A, 34% of *CIC* mutant lung adenocarcinoma carry an additional mutation in *KRAS*. Due to the overall high number of cases with *KRAS* mutations, it is true that the proportion of those cases with *CIC* mutations in turn appears low. However, an analysis of mutual exclusivity using the cBioportal web portal did not reveal statistically significant mutual exclusivity. Moreover, we have further explored the impact of these *CIC* mutations and noticed that almost half of these cases were inactivating mutations including deep deletions, truncating mutations or mutations that affect DNA binding. In a previous publication, we have revealed that most truncating *CIC* mutations lack the C-terminal C1 motif required for DNA binding and repression (PMID: 28278156). In addition, a large proportion of missense mutations affect either the HMG-box or the C1 region, also resulting in defective DNA binding and repression. While the impact of the remaining *CIC* mutations is currently unknown, similar mutations in uncharacterized protein regions indeed resulted in a loss of function, although the mechanisms are unclear (PMID: 27869830). Therefore, we believe that there are indeed patients with an activating *KRAS* mutations and a concomitant inactivating *CIC* mutation. Whether *CIC* mutations are indeed more likely to coincide with *KRAS* mutations in the absence of co-occurring *KRAS* amplifications can only be assessed when significantly more patient data become available.

7. The data shown in Figure 3 using Ad-CC10-CRE to target allelic activation in Club cells do not offer a "likely explanation for the increased tumour burden in KPCic lungs", contrary to what is stated. On the contrary, no actual tumours are shown to arise - only areas of dysplastic tissue that likely represent pseudo-oncogenic dead-ends.

The reviewer raises an important point. To address this, we have now infected a new group of animals (5 KP and 5 KPCic mice) with Ad-CC10-Cre and collected their lungs 4 months after infection. We have then analyzed their tumor burden and could only detect lesions as well as small tumors in the lungs from KPCic mice, indicating that the elevated tumor burden in KPCic animals indeed originated, at least in part, from more effective transformation of Club cells. We only detected one small lesion in 1/5 KP mice. While it is unclear whether this lesion had the potential to progress, we have still included it in our quantification (see Fig. EV4G, upper image). Importantly, all tumors in KPCic mice were significantly larger. We have included the results in the new Figure EV4G and changed the text on page 12 accordingly (see also response to comment 2 by reviewer 2).

Fig. EV4G

8. In the Cic restoration to cell line experiments, the level of GFP-Cic expression relative to endogenous Cic is not discernible from the blots shown in 4C.

The reviewer is correct in pointing this out. We believe that absence of endogenous CIC in this blot can be explained by the relatively high GFP-CIC expression levels driven by our adenoviral vectors. This blot was revealed with an HRP-linked secondary antibody in an Invitrogen iBright scanner and we were not able to detect endogenous CIC under these conditions. However, we have now re-analyzed the protein extracts used for Fig. 4C of the samples infected with Ad-GFP and could readily detect endogenous CIC (see Fig. for reviewer 1). Thus, we concluded that these cells do express endogenous CIC although it cannot be detected due to technical limitations in the context of GFP-CIC overexpression.

Fig. for reviewer 1

9. Please include the Z-score range for Figure 5A

We did not include the Z-score range for this figure because of extraordinary high values of individual datapoints. Thus, we believe it is more informative to represent the respective minima and maxima of each row separately as relative values. Nevertheless, the exact value of each datapoint is shown in Dataset EV3 within the "Heatmap" tab.

10. In Figure 5, Ets proteins remain strikingly sensitive to Trametinib in the KPCic cell lines despite the apparent resilience of Ets transcripts. While depletion of Ets from KPCic lines does restore a degree of sensitivity to Trametinib, in one of the lines depletion of Ets4/5 alone decimates colony formation in the absence of MEK inhibition. Does overexpression of Ets render KP or H358 cells resistant to Trametinib? The data at present do not fully support the headline that Ets mediate resistance. The loss of proliferation in human cells depleted of ETS may be entirely unrelated to CIC status.

The reviewer is correct in pointing this out. The Western blot in Fig. 5C shows that there must be CIC-independent mechanisms that control the stability of ETV4 and ETV5 via potential posttranslational mechanisms. Yet, concomitant silencing of their transcripts is sufficient to revert resistance upon loss of CIC. To address the point raised by the reviewer, we ectopically expressed ETV4 in a KP cell line and observed a clear reduction in the sensitivity to trametinib. Together with a study showing similar findings (PMID: 28178529), our results indicate that high expression levels of ETV4 can promote resistance to trametinib despite potential mechanisms controlling ETV4 stability. We have included these results in the new Fig. EV5E,F and changed the text on page 14 accordingly.

Fig. EV5E,F

11. The final statement of the results section ("...CIC is a crucial effector of KRAS pathway in LuAD") does not make sense.

We agree with the comment made by the reviewer and have therefore eliminated this part of the final sentence on page 19 (end of discussion section).

Referee #2:

1. Figure 2B. I do not find the plot very intuitive. Could there just be a dot for each tumor. I presume that data is per tumor, not per mouse. So N=6 mice etc is not as helpful as presenting N=how many tumors. Actually, I see now that each column is a mouse. In that case the data seems quite weird, why would all tumors in some mice be 2 copies while in other mice the majority of tumors have >3 copies. I guess it's nice to plot per mouse somewhere, but perhaps they can just bundle all tumors across all mice and then make comparisons. This seems like a very important comparison.

We appreciate the comment made by the reviewer and apologize for not having included sufficient information to the figure legend. Indeed, each bar represents an individual tumor. Since these tumors are highly heterogeneous with respect to *Kras* copy numbers, we think it would be informative representing the proportion of cells that show distinct copy numbers within each tumor using violin plots. We have now included the number of tumors analyzed and the corresponding number of mice from which they were derived for each time point to the figure legend: “Quantification of *Kras* copy numbers by FISH in sections from lung tumors in KP mice obtained 1 (n=6 tumors; n=2 mice), 3 (n=10 tumors; n= 2 mice), 5 (n=12 tumors, n=3 mice) and 7 months (n=10 tumors; n=2 mice) after infection with Ad-Cre”. The same applies to Fig. 2E where we have also included this information: “Left, quantification of *Kras* copy numbers by FISH staining in sections from lung tumors in KP (n=12 tumors, n=3 mice; blue symbols) and KPCic (n=17 tumors, n=3 mice, red symbols) mice obtained 5 months after infection with Ad-Cre.”

2. While the authors propose the CIC inactivation enable enhance tumorigenesis from CC10+ cells. I do not see the data to support this. Are there more tumors and/or larger tumors in Ad-CC10-Cre KPCic mice relative to KP mice? There is one tumor shown in Fig 3C, but is there really more big tumors? They should show tumor burden like in Figure . Are they confident that this is THE reason, or could this be ONE reason for the effect of CIC inactivation? Also, they do not provide any evidence the CIC isn't a tumor suppressor when tumor arrive from AT2 cells. Again, I am not saying that they should do that be rather be careful about their claims.

We thank the reviewer for this comment. We agree that the results shown in Fig. 3C are likely only one explanation for the increased tumor burden in KPCic mice. To further substantiate this point, we have now infected a new group of animals (5 KP and 5 KPCic mice) with Ad-CC10-Cre and collected their lungs 4 months after infection. We have then analyzed their tumor burden and could only detect lesions as well as small tumors in the lungs from KPCic mice, indicating that the elevated tumor burden in KPCic animals indeed originated, at least in part, from more effective transformation of Club cells. We only detected one small lesion in 1/5 KP mice. While it is unclear whether this lesion had the potential to progress, we have still included it in our quantification (see Fig. EV4G, upper image). Importantly, all tumors in KPCic mice were significantly larger. We have included the results in the new Figure EV4G and changed the text on page 12 accordingly, including the conclusion: “...could be a likely explanation for their increased tumor burden.” (see also response to comment 7 by reviewer 1).

Fig. EV4G

3. For cell line studies in Figures 4-6 the authors should make sure that all data is from multiple cell lines ($n=3+$ would be best). Maybe they don't need to repeat every experiment but a few more of the critical ones should be repeated in multiple KP and KPCic lines.

All colony assays in Figures 4 and 5 were performed in 2 independent KP as well as KPCic cell lines. Since KP cell lines were not resistant to trametinib, we repeated the experiments now in a third KPCic cell line in which we observed similar results. We have added these data to the new Appendix Fig. S1. We have also modified Figs. 4 and 5 to indicate that these are independent cell lines by labeling them KP1, KP2, KPCic1, and KPCic2. The data shown in Fig. 4A,B were already the average of 3 independent KP and KPCic cell lines.

4. The development of tumors in CIC/p53 mice is a very interesting result as LUADs without driving oncogenes represent a large fraction of lung tumors. I think that this analysis could be improved by providing addition high resolution images, getting assess from a pathologist, maybe HMGA2 staining. The variability across mice is concerning so maybe just some more data to convince the reader that these are really lung tumors (were they able to generate a cell line? I would hope they tried as they are p53-deficient tumors). For the analysis of KRAS mutations they should also look for Q61 mutations, and maybe for BRAF and EGFR mutations. I would not suggest looking harder than that as they likely have other genomic changes but the major point here would be that they lack canonical LUAD oncogenes.

We thank the reviewer for this helpful suggestion, which has been key to prevent an erroneous interpretation of our data. We have now sequenced 3 PCic tumors from FFPE material together with adjacent normal tissue and indeed detected novel latent *Kras* mutations in all three tumors (G12R, Q61R, Q61R). These observations completely changed the interpretation of our results, indicating that combined absence of *Cic* and *Trp53* was not able to induce tumor growth independently of *Kras* mutations. In contrast, their combined absence only enhanced tumor growth upon spontaneous *Kras* mutation, which is probably a rare event that also explains why only 10/22 mice developed tumors. Since this only occurred when both *Cic* and *Trp53* were eliminated, our results nevertheless confirm the data shown in Figures 1 and 2, which indicate that absence of *Cic* and *Trp53* enhances initiation of lung tumors in the absence of *Kras* oncogene amplifications which is indeed also the case here. Therefore, instead of eliminating this figure, we believe that it provides additional evidence for the role of *Cic* loss in initiation of KP-driven LUADs. Thus, we have now re-written this section in the results (pages 15/16) as well as the corresponding part of the discussion (page 18) to indicate our new findings, which finally strengthen our conclusion that: "...our data propose that disruption of *Cic* facilitates initiation of lung adenocarcinomas in KP mice, possibly by rendering *Kras* allelic expansion unnecessary at early stages." (page16)

5. While the impact of CIC inactivation significant, this is nowhere as big of an effect as inactivating many other tumor suppressor (e.g *Lkb1*). In particular the effect of inactivating NF1 (Wang et al. *EmboMM* 2019) is much larger. Thus, it seems unlikely that THE effect of increased MAPK is to inactivate CIC. Not sure if the authors want to mention this.

We agree with the reviewer that inactivation of multiple other tumor suppressors has stronger effects on tumor formation. We believe that the moderate impact of CIC inactivation is likely to stem from derepression of a small subset of KRAS regulated genes, while inactivation of *Nf1* for instance is more likely to amplify all KRAS-regulated pathways (in addition to other pathways such as FAK1). Moreover, based on our new RNAseq data from KP and KPCic tumors (see point 4, reviewer 1), it seems that at the time point analyzed (5 months after infection) both groups are extremely similar, suggesting that the effects of CIC inactivation must be confined to a small window early after infection. We have added a small section to the discussion to discuss the difference to *Nf1* inactivation and speculate on the reason for the mild effect of CIC inactivation, especially in light of our new RNA-seq data (page 18).

6. Stating the "CIC inactivation is a prerequisite for efficient tumor development" is a little too strong. It's just not quite the right word.

The reviewer is right this phrase is a little too strong. Therefore, we have now changed the phrase on page 7 to: "...and reveal that CIC inactivation enhances tumor development." (see also comment 3 by reviewer 1)

7. The conclusion about increased Grade 3 tumors at end point is pretty weak, not sure if they really want to claim that. Most people would present that fraction of tumor of each grade, otherwise it more about tumor number and less about progression. Regardless of what they decide to claim regarding Grade there doesn't really seem to be a difference in progression.

We totally agree with the reviewer that the animal cohort used to determine survival, where we observed a small increase in grade 3 lesions, is not adequate to determine tumor progression. Thus, we have analyzed a more controlled cohort where we analyzed tumor burden 5 months after infection. In this cohort, we detected an increase in the number of lesions upon *Cic* loss (Fig. 1C and Fig. EV1F). Together with the quantification of HMGA2+ tumors (Fig. EV1G) we indeed concluded that CIC inactivation does not affect tumor progression, only tumor initiation. As the reviewer correctly points out, only the number of tumors is different.

8. The IHC in Figure 1E doesn't seem representative of the data that is quantified to the right.

As suggested by the reviewer, we have replaced the IHC image of the KPCic tumor with one that is more representative of the average pERK+ area in KPCic tumors. We now believe that both images represent the average positive areas.

9. The absence of any data comparing K to KCic mice is striking. The authors might want to mention something about this in the Discussion. Whether the impact of CIC inactivation also exists in the KRAS/p53-proficient tumors remains unknown.

The reviewer raises an important issue (see also point 2, reviewer 1 and point 1, reviewer 3). Unfortunately, it was always our intention to perform this study in the context of p53 loss since *Kras*^{G12V} activation in a p53 WT context results in relatively slow tumor progression rates and would have increased the number of required mice considerably. Nevertheless, during the early phases of generating KP and KPCic strains, we obtained a few K as well as KCic animals with WT p53. We infected a low number of these animals (3+3) with Ad-Cre and collected samples when these animals became sick (between 10 and 12 months). We have now used these samples to analyze *Kras* copy numbers by FISH and, although the difference does not quite reach statistical significance ($p=0.0828$, χ^2), there is also a tendency towards higher *Kras* copy numbers in K mice. When only comparing the percentage of tumor cells retaining 2 *Kras* copies, we also observed a trend towards higher percentages in KCic mice ($p=0.09$). However, since statistical significance is not reached, we concluded that absence of CIC does not significantly suppress *Kras* copy number expansion when p53 is retained. These results were included in the manuscript as the new Figs. EV4A+B including a statement in the results section

on page 10 (“...this suppression was not as evident in tumors retaining functional p53.”) and a statement in the discussion section on page 18. There, we reasoned that lack of statistically significant differences could be a consequence of the relative old age of these animals (10-12 months after infection). Moreover, since our results in fact only expose *Cic*'s tumor suppressor activity in the KP model, we propose to change the title of our manuscript to: “*The repressor Capicua is a barrier to lung tumor development driven by KRAS/TRP53 mutations.*”

Figs. EV4A+B

10. If they want to claim that pERK correlates with Kras copy number, then they need to plot on a per tumor level, not just a correlation across time points.

We thank the reviewer for comment. We have now changed the text on page 10 to: “Moreover, MAPK signaling increased concomitantly over time...”, since our data are no direct experimental evidence for *Kras* copy number gains causing elevated pERK levels. Yet, we hypothesize during the manuscript that *Kras* copy number gains could be one explanation.

11. Figure 2G is great. But are 2F and 2G just the same data plotted two different ways? If so, then remove 2F.

Figures 2F and 2G are derived from the same source DNA but analyzed in different ways. As suggested by the reviewer, we have removed Fig. 2F so that Fig. 2G from our original manuscript is now the new Fig. 2F.

12. It really is easier to review manuscripts that have line numbers and the legends on the figures, but I managed.

We apologize to the reviewer, but the instructions for authors in this journal did not specify the inclusion line numbers.

13. The running title doesn't seem quite right.

We agree with the reviewer that the running title was not optimal. Hence, we changed the running title to: “*CIC* deficiency promotes lung tumor initiation.”

Referee #3:

1. Have the authors made *KCic* (no p53 deletion) mice and compared to *K* or *KP* mice for Fig. 1? Based on Fig. 7 data, would *KCic* be similar to *K*? Is *CIC* dispensable as a tumor suppressor for *Kras* mutant lung cancer in the presence of intact p53? It would be good to show the data and/or include in the text.

We agree with the reviewer that is a highly relevant question. Unfortunately, it was always our intention to perform this study in the context of p53 loss since *Kras*^{G12V} activation in a p53 WT context results in relatively slow tumor progression rates and would have increased the number of required mice dramatically. Nevertheless, during the early phases of generating *KP* and *KPCic* strains, we obtained a few *K* as well as *KCic* animals with WT p53. We infected a low number of these animals (3+3) with Adeno-Cre and collected samples when these animals became sick (between 10 and 12 months). We have used these samples to analyze *Kras* copy

numbers by FISH and, although the difference does not quite reach statistical significance ($p=0.0828$, Chi^2), there is also a tendency towards higher *Kras* copy numbers in K mice. When only comparing the percentage of tumor cells retaining 2 *Kras* copies, we also observed a trend towards higher percentages in KCic mice ($p=0.09$). However, since statistical significance is not reached, we concluded that absence of CIC does not significantly suppress *Kras* copy number expansion when p53 is retained. These results were included in the manuscript as the new Figs. EV4A+B including a statement in the results section on page 10 (“...this suppression was not evident in tumors retaining functional p53.”) and a statement in the discussion section on page. Moreover, since our results in fact only expose *Cic*'s tumor suppressor activity in the KP model, we propose to change the title of our manuscript to: “*The repressor Capicua is a barrier to lung tumor development driven by Kras/Trp52 mutations.*” (see also point 2, reviewer 1 and point 9, reviewer 2).

Figs. EV4A+B

2. Etv4 and Etv5 as downstream targets for CIC have been reported by others (Okimoto 2017 and Park 2017). It would be better to report new genes regulated by CIC that are critical for lung tumor growth. RNA-seq using the samples from Fig 1G may help. The authors should also do ChIP-seq instead of ChIP to demonstrate a better mechanism by which CIC regulates downstream genes. Ccnd1 mRNA data in Fig. 1G doesn't seem to be consistent with ChIP data at Ccnd1 locus in Fig. S2C. RNA-seq and ChIP-seq will provide a better and unbiased understanding of CIC as a transcriptional repressor and the lung cancer field would appreciate the effort.

We agree with the reviewer that *Etv4* and *Etv5* have been reported as some of the most studied CIC target genes. Yet, we detected these genes in an unbiased assay to detect potential drivers of resistance (Fig. 5A). These observations, in alignment with several other studies (e.g., PMID: 28178529) suggest that CIC exerts its tumor suppressive effects mostly via repression of *Etv4* and *Etv5*. As further suggested, be performed ChIP-seq experiments to identify promoters to which CIC can bind upon inhibition of the MAPK pathway. This analysis confirmed several known CIC target genes, including *Etv4* and *Etv5*, negative regulators of MAPK signaling, or *Ccnd1*. Since we also performed a new RNA-seq experiment of KP and KPCic lung tumors, the old Fig. 1G became obsolete and has been removed from the manuscript. Yet, we could not identify any differentially expressed CIC target gene (new Fig. 1G), suggesting that the degree of CIC inactivation is similar in KP and KPCic tumors, at least at the time point analyzed (5 months). It will definitely be interesting to explore at the transcriptome of KP and KPCic tumors at earlier time points, especially when *Kras* is not yet amplified, to determine the impact of *Cic* inactivation on tumor initiation. However, this will be the subject of future studies. Thus, we concluded that CIC inactivation exerts no additional effect beyond KRAS activation in established tumors or cell lines. However, in the absence of CIC, some of these genes (mostly *Etv4* and *Etv5*) remain active upon inhibition of the MAPK pathway and promote resistance to MAPK pathway inhibitors. We have included these data to the new Fig. 1G as well as the new Fig. EV2. We also included the ChIP-seq (Dataset EV1) and RNA-seq (Dataset EV2) datasets to the manuscript and changed the text on page 9 (results) and page 18 (discussion) accordingly. Nevertheless, our ChIP-seq analysis has indeed revealed several poorly described CIC target genes that merit further investigation in future studies.

3. For Fig. 3C, I recommend looking at SOX2 expression to ensure the author's claim that Cic deletion accelerates the transformation of Club cells into ATII-like tumor cells. I wonder if SOX2 expression is maintained or lost in the KPCic tumors. SOX2 staining information is available at Sutherland et al (2014).

We thank the reviewer for this recommendation. We have now performed SOX2 IHCs in sections from KP or KPCic tumors 5 months after infection with Ad-CC10-Cre and observed that SOX2 expression was largely maintained. Representative images have been included in the new Fig. EV4F. Moreover, we have described this result on page 12 in the results section.

Fig. EV4F

4. In Fig. 4, does CIC or CICS173A function without ATXN1L in the in vitro cell line experiments? The authors should show the data as to the function of CIC and CICS173A without ATXN1L in the in vitro cell line. Did these cell lines lose ATXN1L after they were isolated from lung tumors? If so, it's good to show the data (RNA and/or protein) as well (in vitro vs in vivo) and mention these data in the text.

As suggested by the reviewer, we have now analyzed the expression levels of *Atxn1* and *Atxn1l* both in KP and KPCic tumors as well as in corresponding cell lines by qRT-PCR due to the lack of reliable antibodies. Indeed, KP and KPCic cell lines expressed significantly less *Atxn1* and *Atxn1l* mRNA than the respective tumors, thus suggesting that expression levels became reduced upon establishment of cell cultures (new Fig. EV5B). Moreover, we show that for efficient gene repression, increasing the levels of one of the co-repressors – we ectopically expressed ATXN1L – is necessary (new Fig. EV5C). We believe that this stems from a combination of reduced expression levels in cell lines together with relatively high levels of CIC overexpression after adenoviral transduction, resulting in a profound disbalance between CIC and its co-repressors.

Figs. EV5B+C

1st Sep 2025

Dear Dr. Drosten,

Thank you for submitting your revised study, that was reviewed by the three initial referees. As you will see below, they are overall satisfied with the revisions, and I will therefore be able to accept your manuscript once the following editorial concerns are addressed:

1/ Referee's concerns:

Please address the remaining minor concerns raised by referee #3, experimentally or in writing.

2/ Manuscript text:

- Please remove the red font text and only keep in track changes mode any new modification in the text.
- "Conflict of interest disclosure statement" should be renamed "Disclosure and conflict of Interests Statement" and moved after the Acknowledgements.
- "Materials and Methods" should be renamed "Methods":
 - o Animals: Please provide the reference number for approval. Please indicate the age of the mice at time of experiments.
 - o Cells: please indicate whether the cells were authenticated and tested for mycoplasma contamination.
 - o Antibodies: please provide dilutions/concentrations.
 - o Statistics: please provide a statement on sample randomization.
- "Data availability" should be placed after "Statistical analyses". Please note that the datasets should be publicly available before acceptance of the manuscript. Please provide URLs for all deposited datasets.
- Acknowledgements: please note that the information provided should match the information entered in the submission system (currently, Spanish National Research and Development Plan, Instituto de Salud Carlos III, LABAE20049RODR, P2022/BMD-7321/MITIC-CM, ERC-AG/695566, THERACAN, and any of the fellowships or contracts mentioned are not listed as funders in our system).
- The heading "Extra View Figures" should be corrected to "Expanded View Figure Legends".

3/ Figures:

- There is a callout for a Table EV1 but not such table was provided, please correct; a callout is missing in the manuscript text for Fig 7C.
- Please upload your Appendix file as a PDF.
- Figures/figure panels may be re-used, however it must be indicated in the figure legends (i.e. fig 1D and EV1H).
- Please address the queries from our data editors in the figure legends:
 1. Please indicate the statistical test used for data analysis in the legends of figures 6A, EV2 C.
 2. Please note that the error bars are not defined in the legends of figures 5H, 6C.
 3. Please note that the dotted borders are not defined in the legend of figure 2A. This needs to be rectified.

4/ Thank you for providing Source Data. For Figure 1D, please provide Source Data for all images and carefully check the labeling of the provided individual images (i.e. KP_CC10).

5/ The paper explained: I introduced minor edits in your text, please let us know if you agree or amend as you see fit, and include the text in the main manuscript file:

Problem

Lung adenocarcinomas driven by KRAS oncogenes are among the most severe and lethal types of cancer. Yet, the mechanisms linking KRAS mutations and progressive allelic imbalances to tumor formation remain unclear.

Results

Our study shows that mutant Kras copy numbers gradually increase during tumor progression. Genetic disruption of the repressor Capicua (CIC) in Kras/Trp53 mutant mice suppresses these allelic imbalances and promotes the transformation of bronchiolar Club cells, leading to an increased tumor burden and inducing resistance to MAPK pathway inhibition. Restoring CIC repressor activity or silencing of its target genes Etv4 and Etv5 decreases resistance to MAPK pathway inhibition, similar to treatment with drugs (PFK15 and Tx-1123) that selectively affect the viability of Cic-deficient tumor cells.

Impact

We have identified the repressor CIC as a barrier to lung tumor development in Kras/Trp53 mutant mice. The absence or mutational inactivation of CIC in lung cancer patients may have adverse consequences such as increased resistance to MAPK pathway inhibitors, and reveals new vulnerabilities that could be exploited to overcome resistance.

6/ I introduced minor edits in your synopsis text, please let me know if you agree or amend as you see fit:

"The inactivation of the repressor Capicua (CIC) promotes the development of lung cancer in Kras/Trp53 mutant mice by transforming bronchiolar club cells. Loss of CIC in lung cancer cells causes resistance to MAPK pathway inhibition while exposing new drug vulnerabilities.

- Absence of CIC abrogates the requirement for Kras oncogene amplifications in Kras/Trp53 mutant mice during tumor development.
- CIC deficiency promotes transformation of bronchiolar Club cells.
- Silencing of CIC target genes ETV4 and ETV5 restores the sensitivity to the MEK inhibitor trametinib in Cic-deficient tumor cells
- Lung tumor cells lacking Cic display increased sensitivity to PFK15 and Tx-1123."

Thank you for providing a nice visual abstract. I have cropped a small portion of this image to serve as a thumbnail for the table of content on our webpage (attached). Please let us know if you agree with the selection, or provide a different one at the same dimensions.

7/ As part of the EMBO Publications transparent editorial process initiative (see our Editorial at <http://embomolmed.embopress.org/content/2/9/329>), EMBO Molecular Medicine will publish online a Review Process File (RPF) to accompany accepted manuscripts.

This file will be published in conjunction with your paper and will include the anonymous referee reports, your point-by-point response and all pertinent correspondence relating to the manuscript. Let us know whether you agree with the publication of the RPF and as here, if you want to remove or not any figures from it prior to publication. Please note that the Authors checklist will be published at the end of the RPF.

I look forward to receiving your revised manuscript.

Yours sincerely,

Lise Roth

***** Reviewer's comments *****

Referee #1 (Remarks for Author):

The authors have addressed the issues I raised at first review to my satisfaction - I am happy to recommend acceptance for publication.

Referee #2 (Remarks for Author):

The authors have address my concerns and those of the other Reviewers well. I congratulate them on this interesting work.

Referee #3 (Remarks for Author):

The authors addressed my comments; however, I have a few minor suggestions that the authors should be able to address in a short time frame.

1) Since the title has changed from "...by KRAS oncogenes" to "...Kras/Trp53 mutations" based on the new mouse data, the authors should relate their new findings in mice to human lung adenocarcinoma cases that carry KRAS mutations/TP53 mutations/CIC mutations instead of TP53 mutations/CIC mutations or KRAS mutations/CIC mutations (Fig. 7), which will provide more human relevance to the authors' study.

2) In the previous comment, I asked whether CIC or CICS173A functions without ATXN1L in the in vitro cell line experiments and suggested that the authors should show the data as to the function of CIC and CICS173A without ATXN1L in the in vitro cell line. The cell proliferation and viability data in Fig. 4A and 4B do not have ectopic ATXN1L while the other gene expression and colony formation data in Fig. 4C-4F have ectopic ATXN1L. The newly added data on the expression of Etv4 in KPCic cells in Fig. EV5C has both with and without ectopic ATXN1L. These data indicate that the authors are capable of showing the cell proliferation and viability data of KP and KPCic cells with ectopic ATXN1L as well, which will make all data in Fig. 4 with ectopic ATXN1L that is supposed to mimic the in vivo data better according to the author's new data in Fig. EV5B. Since ATXN1L appeared only in Fig. 4 for the in vitro experiments, it would be good to have a consistent in vitro dataset with ectopic ATXN1L at least in Fig. 4 (or all without ectopic ATXN1L if the authors want to be consistent with other in vitro experiment data). And it would be good to clarify in the text the reason why the authors did not have to add ectopic ATXN1L in other in vitro experiments in Fig. 5, 6 and other supplementary figures.

3) The authors' effort on ChIP-seq, showing that CIC binds to the loci of Etv4 and Etv5 in KP cells with trametinib, is great. However, the ChIP-seq in the KP and KPCic cell lines was conducted without ectopic ATXN1L. It would be good to mention the absence of the ectopic ATXN1L in this ChIP-seq study in the text as the reason was already discussed in Comment 2 above.

Response to reviewers:

We thank referee #3 for these additional suggestions to improve our manuscript. Below, we provide a point-by-point response to his/her comments. We also slightly changed the order of the figures to improve the clarity of our manuscript. In particular, we moved the previous figure 7 right after figure 3, now becoming the new figure 4. The other figures were re-numbered accordingly. Due to the new results obtained during the first round of revision, we now felt that this figure connected better with the first part of the paper (in vivo data). We also moved the corresponding part of the text to its new position without changing the text.

Referee #3:

1. Since the title has changed from "...by KRAS oncogenes" to "...Kras/Trp53 mutations" based on the new mouse data, the authors should relate their new findings in mice to human lung adenocarcinoma cases that carry KRAS mutations/TP53 mutations/CIC mutations instead of TP53 mutations/CIC mutations or KRAS mutations/CIC mutations (Fig. 7), which will provide more human relevance to the authors' study.

We thank the reviewer for this comment. We agree that a correlation of our mouse data to human *KRAS/TP53/CIC* mutant lung adenocarcinomas would be very informative. Following this suggestion, we now compared these with *KRAS/TP53* mutant tumors, as we have done using mouse models, despite identifying only a small number LUAD patients with *KRAS/TP53/CIC* mutations in the TCGA database. We did not observe differences in expression of MAPK pathway regulated genes including *CIC* target genes. However, we did detect several differentially expressed genes, suggesting that these tumors might not be completely identical to *KRAS/TP53* mutant tumors. However, the low number of concurrent *KRAS/TP53/CIC* mutations does not allow a meaningful correlation of this mutational profile with clinical parameters. Thus, we have now eliminated the previous Appendix Figure S4 and added a new Appendix Figure S1 showing these data. We have also included a description of these new results on page 13. In conclusion, this analysis shows that combined *KRAS/TP53/CIC* mutations exist in human lung adenocarcinoma patients, but more samples are required until clinically relevant conclusions can be drawn.

Fig. S4

New Appendix Fig. S4

2. In the previous comment, I asked whether CIC or CICS173A functions without ATXN1L in the *in vitro* cell line experiments and suggested that the authors should show the data as to the function of CIC and CICS173A without ATXN1L in the *in vitro* cell line. The cell proliferation and viability data in Fig. 4A and 4B do not have ectopic ATXN1L while the other gene expression and colony formation data in Fig. 4C-4F have ectopic ATXN1L. The newly added data on the expression of Etv4 in KPCic cells in Fig. EV5C has both with and without ectopic ATXN1L. These data indicate that the authors are capable of showing the cell proliferation and viability data of KP and KPCic cells with ectopic ATXN1L as well, which will make all data in Fig. 4 with ectopic ATXN1L that is supposed to mimic the *in vivo* data better according to the author's new data in Fig. EV5B. Since ATXN1L appeared only in Fig. 4 for the *in vitro* experiments, it would be good to have a consistent *in vitro* dataset with ectopic ATXN1L at least in Fig. 4 (or all without ectopic ATXN1L if the authors want to be consistent with other *in vitro* experiment data). And it would be good to clarify in the text the reason why the authors did not have to add ectopic ATXN1L in other *in vitro* experiments in Fig. 5, 6 and other supplementary figures.

We perfectly understand the concern raised by the reviewer. However, our data (especially those shown in the original Figs. 4B, 5A and 5B, now Figs. 5B, 6A and 6B) show that endogenous

Atxn1/Atxn1l levels must still be sufficient to sustain full CIC repressor activity in KP cell lines. This is especially reflected in the near complete repression of *Etv4* and *Etv5*, two of the most robust CIC targets genes found in our ChIP-seq data and multiple other studies, whose mRNA expression levels seem to fully depend on CIC repressor activity. Therefore, we believe that ectopic ATXN1L expression in Figs. 4A and B (now Figs. 5A and B) would introduce an unnecessary variable. For instance, trametinib resistance in KPCic cells must be a direct consequence of CIC loss-of-function, since KP cells retain sensitivity to trametinib, again indicating that the ATXN1/ATXN1L/CIC complex must remain fully functional in the latter (Fig. 4B). Since adenoviral overexpression of WT or mutant CIC results in extremely high expression levels, reconstitution of CIC repressor activity in this specific context required concomitant overexpression of ATXN1L, while not required in all other scenarios. Moreover, as shown in Fig. 1 for Reviewers below, ectopic expression of ATXN1L does not affect proliferation or trametinib sensitivity in KP cells.

Figure 1 for Reviewers: Comparison between KP and KP-ATXN1L cell lines. (A) Proliferation in 5% FBS. (B) Proliferation in 10% FBS. (C) Relative viability after treatment with the indicated concentrations of trametinib for 72h.

For the reasons outlined above and together with the data shown in Figure 1 for Reviewers, we believe that it is unnecessary to add ATXN1L to Figs. 4A and B (now Figs. 5A and B). Nevertheless, to better explain why we did not have to add ectopic ATXN1L, as proposed by the reviewer, we have changed the text on page 14:

“Moreover, to efficiently reconstitute CIC’s transcriptional repressor activity, we ectopically expressed the CIC corepressor ATXN1L in these experiments...”

and on page 15:

*“These results also indicate that, although endogenous *Atxn1/Atxn1l* levels are significantly reduced, they remain sufficient to sustain CIC repressor activity in KP cell lines without the need for ectopic ATXN1L expression.”*

We have also modified a section in the discussion section on page 20:

*“Interestingly, reconstitution of CIC repressor activity via ectopic expression of CIC^{S173A}, a mutant CIC protein that is less responsive to ERK-mediated inactivation, and ATXN1L, a co-repressor required for CIC stability and DNA binding (Lee et al, 2011; Wong et al, 2019), not only reduced tumor cell proliferation, but also restored sensitivity to trametinib. Concomitant ATXN1L overexpression was only required upon adenoviral expression of CIC variants, since endogenous *Atxn1/Atxn1l* levels, albeit reduced, were sufficient to sustain CIC’s repressor activity in KP cells.”*

3) *The authors' effort on ChIP-seq, showing that CIC binds to the loci of Etv4 and Etv5 in KP cells with trametinib, is great. However, the ChIP-seq in the KP and KPCic cell lines was conducted without ectopic ATXN1L. It would be good to mention the absence of the ectopic ATXN1L in this ChIP-seq study in the text as the reason was already discussed in Comment 2 above.*

Our ChIP-seq data revealed that endogenous CIC binds to its target promoters with high efficiency upon treatment with trametinib. Since, as described in Comment 2, these binding sites coincide with the genes repressed upon trametinib treatment in KP but not in KPCic cells (Figs. 6A and B), we concluded that CIC is still capable of binding to DNA and repressing its target genes in KP cells, despite a reduction in endogenous *Atxn1/Atxn1l* levels. Therefore, we believe that the reasons outlined in Comment 2 also apply here. Given that the CIC repressor complex is still functional, there is no requirement for ectopic expression of ATXN1L. Thus, the sentence added to page 15 also explains why we did not have to add ATXN1L for any other experiment including ChIP-seq:

*“These results also indicate that, although endogenous *Atxn1/Atxn1l* levels are significantly reduced, they remain sufficient to sustain CIC repressor activity in KP cell lines without the need for ectopic ATXN1L expression.”*

9th Oct 2025

Dear Dr. Drosten,

Thank you for submitting your revised files. I am pleased to inform you that your manuscript is accepted for publication and is now being sent to our publisher to be included in the next available issue of EMBO Molecular Medicine.

With kind regards,

Lise Roth
